# On the Limits of LLM Adaptability: Impact of Model-Internalized Priors on Annotation Task Performance

Etienne Casanova [* 1]   Rafal Kocielnik [* 1]   R. Michael Alvarez [1]

## Abstract

Large Language Models (LLMs) are increasingly used for zero-shot annotation and LLM-as-a-judge tasks, yet their reliability hinges on how model-internalized priors interact with user-provided instructions. We investigate three dimensions of this interaction: (1) how an LLM's familiarity with data and task definitions affects performance, (2) the extent to which additional information in prompts can correct zero-shot errors ("decision stickiness"), and (3) model susceptibility to misaligned task definitions. Through experiments on toxicity detection across diverse datasets (spanning social media, gaming, news, and forums) using both dense and mixture-of-experts models, we find that nearly two-thirds of zero-shot errors are resistant to correction, with an overall rescue rate (fraction of initial errors corrected by prompting) of only 34.8%. High-confidence errors prove especially resistant to correction. When given misaligned definitions, LLMs follow them while maintaining confidence levels unchanged from the aligned condition. Crucially, we introduce Definition-Specific Familiarity (DSF), which measures alignment between a model's internal concept and the task definition. After controlling for dataset-level confounds, DSF shows a positive association with model performance (partial $r = +0.41$), while three distinct memorization metrics (ROUGE-L, BERTScore, and embedding cosine similarity) all fail to show a positive association. These findings show the limitations of prompt-based correction in annotation tasks, highlighting the importance of definition alignment over text-level memorization.

---

[*]Equal contribution. [1]California Institute of Technology, Pasadena, CA, USA. Correspondence to: Etienne Casanova <ecasanov@caltech.edu>, Rafal Kocielnik <rafalko@caltech.edu>.

*Proceedings of the 43rd International Conference on Machine Learning*, Seoul, South Korea. PMLR 306, 2026. Copyright 2026 by the author(s).

## 1. Introduction

Large Language Models (LLMs) are increasingly deployed for zero-shot annotation (Gilardi et al., 2023) and rubric-based evaluation (LLM-as-a-judge) tasks (Zheng et al., 2023; Liu et al., 2023). These use cases are attractive: users provide a task definition via prompt, and the model annotates at scale without labeled training data (Gilardi et al., 2023; Brown et al., 2020). The implicit assumption is that the user's definition governs the model's behavior.

But LLMs are not blank slates. Through pre-training on web-scale corpora (Brown et al., 2020), followed by instruction tuning and reinforcement learning from human feedback (RLHF), models develop implicit understandings of common concepts and tasks. When a user asks an LLM to detect "toxic" content, the model has already been shaped by millions of documents and human-rated examples that discuss, define, and exemplify toxicity, and this exposure shapes an internal concept of the task. Critically, this internalized concept may not align with the user's intended definition: toxicity is operationalized differently across applications, as identity attacks (Borkan et al., 2019), disruptive behavior in online gaming (Kocielnik et al., 2024; 2025b), or as speech targeting protected groups (ElSherief et al., 2018). An LLM anchored to a particular understanding may not simultaneously match all of these definitions, and its internalized priors may constrain how much user instructions can steer its behavior and alignment with the user's intended interpretation (Min et al., 2022). This disconnect between a model's internalized concepts and user-specified definitions represents a fundamental alignment challenge.

When the user's task definition conflicts with the model's internalized concept, which one governs behavior? Prior work has shown that LLMs exhibit biases reflecting the demographics and viewpoints in their training data — both at the pre-training stage and after instruction tuning or RLHF (Nadeem et al., 2021; Nangia et al., 2020; Parrish et al., 2022; Santurkar et al., 2023). Prompting and reasoning techniques can inadvertently amplify such latent biases (Shaikh et al., 2023). Less is known about the degree to which users can override a model's internalized task understanding through prompting, or how to measure the alignment between a model's internal concept and a task definition.

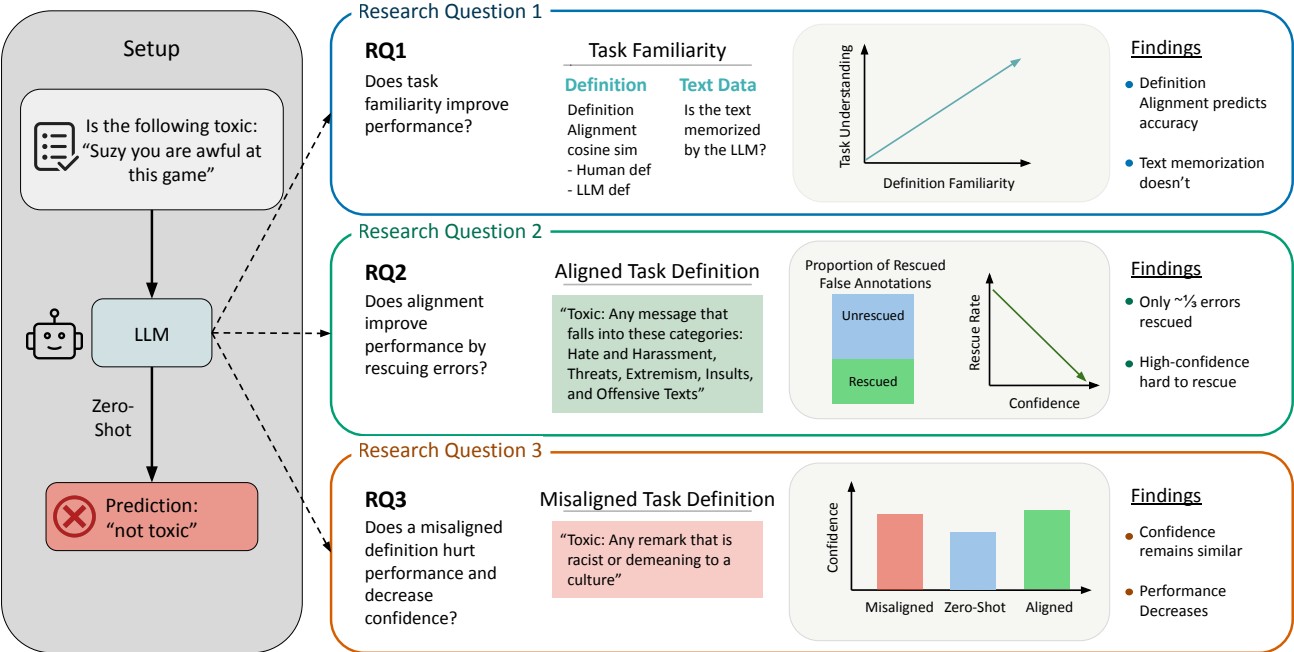

Figure 1. **Study overview and research questions.** *Left:* zero-shot annotation setup: given an input text and a user-provided task definition, the LLM produces a label prediction. *Right:* we study three facets of the interaction between model-internalized task concepts and user instructions. **RQ1** tests whether task familiarity correlates with performance, contrasting *definition alignment* (Definition-Specific Familiarity; DSF) with *text memorization* metrics (ROUGE-L, BERTScore, embedding similarity; Table 4). **RQ2** measures steerability under an aligned definition by asking whether zero-shot errors can be *rescued* (rescue rate), and how this depends on confidence. **RQ3** probes susceptibility to *misaligned* definitions, testing how performance and confidence change when the provided definition is incorrect. Insets summarize the main empirical findings for each question.

We investigate this tension through three research questions (Figure 1):

- **RQ1 (Familiarity and Performance):** How does an LLM's familiarity with the data and task definition affect annotation performance?
- **RQ2 (Decision Stickiness):** To what extent can external knowledge or specialized prompting correct initial zero-shot errors, and how does model confidence relate to correctability?
- **RQ3 (Misalignment Susceptibility):** How do LLMs behave when given misaligned or incorrect task definitions, and can confidence scores detect such misalignment?

To answer these questions, we evaluate 9 LLMs across 5 toxicity detection datasets spanning social media, gaming, news, and forum domains. We focus on toxicity due to the high potential for interpretive tension between LLMs and human task-specific definitions. We introduce *Definition Specific Familiarity (DSF)*, a metric that quantifies alignment between a model's internal concept of a task and the dataset's specific definition. Our key findings are: (1) nearly two-thirds of zero-shot errors resist correction through any prompting strategy, including automated prompt refinement techniques, with an overall rescue rate of only 34.8%; (2) high-confidence errors are especially resistant to correction,

exhibiting strong "decision stickiness"; (3) after controlling for dataset-level confounds, DSF (computed as the consensus across six sentence encoders, on an extended $N = 54$ panel) is positively associated with which models perform better at annotation (partial $r = +0.41$), while text memorization shows no positive association (partial $r = -0.19$); and (4) LLMs faithfully follow misaligned definitions yet remain highly confident even when applying incorrect instructions, revealing a critical calibration failure that prevents confidence-based detection of definition errors. These results suggest that for annotation tasks, conceptual alignment with task definitions, not data familiarity, determines model performance, and task concepts that the model has internalized through training and tuning anchor its behavior in ways that limit the effectiveness of prompt-based steering. Importantly, we treat prompting as the *control channel* and measure fundamental limits of that channel set by these internalized priors. This complements recent work on steerability in generation tasks (Chang et al., 2026). We find that our metric, DSF, is positively associated with which models perform better on a given task, and show that confidence scores provide no reliable signal for detecting when models are applying incorrect definitions. Code and analysis pipelines are publicly available at https://github.com/etmaca5/llm-interna lized-priors-for-annotation.

## 2. Background and Related Work

**LLM Steerability and Controllability.** Recent work has formalized steerability as the degree to which a model's behavior can be shifted from its baseline via interventions such as prompting (Miehling et al., 2025). Chang et al. (2026) introduce a multi-dimensional goal-space framework for measuring steerability in text generation, identifying three key failure modes: poor coverage (rare goals underrepresented), miscalibration (overshooting requests), and side effects (unintended changes to unspecified dimensions). Their evaluation reveals that current LLMs struggle with reliable steerability even under various interventions including prompt engineering and reinforcement learning. Our work addresses a complementary question: in *annotation* tasks where the output is a discrete label rather than generated text, what determines whether a model can be steered toward a specific task definition? We find analogous failure modes: limited reachability (stubborn errors), miscalibration (flat confidence under misalignment), and side effects (rescue-corruption tradeoffs).

**LLMs as Annotators and Judges.** The use of LLMs for evaluating other models or annotating datasets has become a standard paradigm. Gilardi et al. (2023) showed that ChatGPT can outperform crowd-workers on text-annotation tasks, while Zheng et al. (2023) established LLM-as-a-judge evaluation with MT-Bench. Kim et al. (2024) introduced Prometheus, a model specifically fine-tuned for fine-grained rubric-based evaluation, highlighting the importance of specialized alignment for assessment tasks. However, Baumann et al. (2025) replicated 37 annotation tasks from 21 published studies using 18 different language models, finding that configuration choices can lead to incorrect conclusions in 31-50% of cases. Relatedly, Kocielnik et al. (2025a) show that aligning a model's concept with a curated human definition is a prerequisite for reliable large-scale annotation of player chat, motivating explicit definition-alignment diagnostics. With just a few prompt paraphrases across multiple LLMs, researchers can present most hypotheses as statistically significant - a phenomenon coined "LLM hacking." This suggests that LLMs should be treated as complex instruments requiring rigorous validation rather than convenient black-box annotators.

**Model-Internalized Priors and Annotation Reliability.** Modern LLMs develop strong internal priors across the full training pipeline (pre-training, instruction tuning, and RLHF/DPO) that can conflict with user-provided definitions. Carlini et al. (2021) demonstrate that LLMs memorize specific training examples, suggesting models have ingrained conceptions of common concepts like toxicity. Post-training alignment can amplify these effects: Zhang et al. (2026) identify *typicality bias* where annotators systematically fa-

vor familiar text during RLHF, causing models to anchor to "typical" examples rather than following nuanced definitions. Min et al. (2022) show that in-context learning is often driven by format and label space rather than input-label mappings, suggesting models may not fully override their priors even with explicit examples. Prior work on benchmark contamination has developed metrics like Min-K% Prob (Shi et al., 2024), BERTScore (Zhang et al., 2020), and continuation-based detection (Golchin & Surdeanu, 2024), but less is known about how internalized *conceptual* priors (vs. text memorization) constrain steerability.

**Confidence and Calibration.** Model accuracy alone can mask calibration failures critical for annotation reliability. Surveys of LLM confidence estimation (Geng et al., 2024) highlight challenges with overconfidence and unreliable uncertainty quantification. Black-box confidence elicitation methods reveal that current LLMs lack an internally coherent sense of confidence (Han et al., 2025), often overstating self-reported scores while showing wide variability when prompted to reconsider (Xiong et al., 2024; Pawitan & Holmes, 2025). While verbalized confidence can be better calibrated than token probabilities under RLHF (Tian et al., 2023), prompt design strongly shapes reliability (Yang et al., 2024). We adopt verbalized confidence to enable applicability to closed-source models that do not expose token-level probability scores.

## 3. Methods

We propose a framework to evaluate the interplay between the priors an LLM has internalized through training and tuning (familiarity) and its steerability via prompting (alignment) in the context of annotation tasks.

### 3.1. Familiarity Metrics.

We distinguish between two types of familiarity that could influence annotation performance: *text familiarity* (whether the model has memorized specific texts from the dataset) and *definition familiarity* (whether the model's internal concept aligns with the task definition). These represent different mechanisms: text familiarity suggests performance via generation of memorized content, while definition familiarity suggests performance via conceptual task alignment.

**Text Familiarity.** Measures memorization by prompting the model to generate a continuation given a prefix comprising 40% of the text instance, then computing **ROUGE-L F1** (based on longest common subsequence) (Lin, 2004) between the generated continuation and the remaining ground truth text. This adapts the continuation-based contamination detection approach of Golchin & Surdeanu (2024) for continuous familiarity measurement: rather than using dataset-

specific prompts for binary contamination detection, we use generic continuation prompts (see Appendix B) at temperatures 0.0 and 0.7, taking the maximum score. To ensure findings are robust to the choice of memorization proxy, we additionally compute **BERTScore F1** (Zhang et al., 2020) using DeBERTa-XL contextual token embeddings, and **embedding cosine similarity** using the same sentence encoders as DSF—so any difference between metrics reflects signal type (lexical vs. contextual vs. semantic reproduction) rather than encoder choice. High scores indicate that the model can reproduce text that it likely encountered during training.

**Definition-Specific Familiarity (DSF).** Quantifies alignment between a model's internal understanding of the target phenomenon (e.g., what constitutes toxicity) and the dataset's operational definition of that phenomenon. The intuition is analogous to human annotators: two annotators given the same definition may interpret it differently based on their prior conceptions and identities (Sap et al., 2022; Plank, 2022). An annotator whose mental model aligns with the definition will perform better than one with a mismatched conception, regardless of whether they have seen the specific texts before.

To compute DSF, we prompt the model to explain its understanding of the target concept (e.g., "In your own words, what makes content toxic?") and measure semantic similarity between this explanation and the dataset's full definition using sentence embeddings. To remove sensitivity to any single embedding model, we compute DSF as the unweighted mean (*consensus DSF*) across six diverse sentence encoders: MiniLM, MPNet, BGE-large, E5-large, Instructor-large, and OpenAI's `text-embedding-3-small`. Appendix E.1 shows that all six encoders produce positive partial correlations with accuracy and that pairwise DSF values are highly concordant ($r \geq 0.88$). DSF–accuracy correlations are computed across all nine models and six datasets (adding the Jigsaw Unintended Bias dataset (Borkan et al., 2019) to the five primary datasets), yielding $N = 54$ model–dataset pairs. Crucially, DSF can be measured using only the task definition and a handful of prompts, without requiring labeled data or full annotation runs, making it a practical diagnostic for selecting model–task pairings. This quantification captures the entire concept boundary including both positive and negative criteria. We also tested alternative variants incorporating domain-specific prompts and separate positive/negative class definitions, but these showed weaker correlation with model performance on a given task (see Appendix E).

### 3.2. Steerability Metrics

To quantify an LLM's ability to correct its errors when provided with better instructions (steerability), we define

```
[Base classification prompt...]
Respond in exactly this format:
PREDICTION: [LABEL/not LABEL]
CONFIDENCE: [0-100]
```

*Figure 2.* Confidence elicitation prompt suffix appended to all classification prompts. `LABEL` is replaced with the dataset-specific positive class (e.g., `toxic`, `hateful`, `offensive`, etc.). The base prompt varies by condition (zero-shot, definition, few-shot, misaligned, etc.). This template is identical across all models.

the **Rescue Rate**:

$$\text{Rescue Rate} = P(\text{Correct} \mid \text{Prompted}, \text{Zero-Shot Wrong}) \quad (1)$$

This metric captures the reachability of correct behaviors via prompting. A low rescue rate indicates that many behaviors are unreachable through prompt-based steering alone. We define **decision stickiness** as the tendency for high-confidence errors to resist correction.

To investigate how LLMs respond to incorrect instructions (RQ3), we employ *concept substitution*: providing a definition from a different dataset that operationalizes a related but semantically distinct concept. For example, when annotating a general toxicity dataset, we substitute definitions of hate speech (which requires identity-based targeting) for gaming toxicity (which includes any disruptive behavior). We test 6 misaligned conditions varying in *definition scope*: narrow definitions like hate speech require identity-based targeting, while broad definitions like gaming toxicity capture any disruptive behavior. Full definitions are in Appendix A.

We measure four complementary metrics that decompose how prompting changes behavior relative to zero-shot: **Change Rate** (fraction of flipped labels) captures how responsive a model is to the provided definition. **Rescue Rate** ($P(\text{correct} \mid \text{zero-shot wrong})$) quantifies how often prompting fixes errors. **Corruption Rate** ($P(\text{wrong} \mid \text{zero-shot correct})$) measures how often prompting breaks correct predictions. Models that are easily steered can be steered in both beneficial and harmful directions. Finally, **Prediction Bias** (predicted positive rate - true positive rate) captures whether prompting shifts predictions toward overcalling (more false positives) or under-calling (more false negatives) relative to the dataset's base rate.

**Confidence.** We elicit verbalized confidence scores by prompting the model to output a numerical score (0–100) alongside its prediction. The exact prompt suffix is shown in Figure 2. We use the same template across all models; only the label string varies by dataset. All calls use temperature 0.

### 3.3. Experimental Setup

**Models.** We evaluate nine instruction-tuned models spanning open-weights and proprietary systems (Table 2), or-

*Table 1.* Overview of datasets used in our experiments. % Pos. indicates the proportion of positive (toxic/offensive) labels in the original dataset. † only used in supplementary analysis.

| Dataset | Domain | Size | % Pos. |
|---|---|---|---|
| Twitter Hate (Davidson et al., 2017) | Social Media | 24,783 | 5.8 |
| OLID (Zampieri et al., 2019) | Social Media | 14,100 | 33.2 |
| GameTox (Naseem et al., 2025) | Gaming | 53,000 | 42.5 |
| Fox News (Gao & Huang, 2017) | News | 1,528 | 28.5 |
| Jigsaw Toxic Comments (cjadams et al., 2018) | Forum | 159,571 | 9.6 |
| Jigsaw Unintended Bias† (Borkan et al., 2019) | Forum | 97,320 | 8.0 |
| SemEval-2018 Irony† (Van Hee et al., 2018) | Social Media | 4,618 | 48.1 |
| Subjectivity† (Pang & Lee, 2004) | Reviews | 10,000 | 50.0 |

*Table 2.* Overview of models evaluated. Act. = parameters active per forward pass (MoE models). N/D = not disclosed. *Estimated from unofficial sources; not officially disclosed by OpenAI.

| Model | Arch. | Param. | Active |
|---|---|---|---|
| Llama-3.1-8B (Grattafiori et al., 2024) | Dense | 8B | 8B |
| Llama-3.1-70B (Grattafiori et al., 2024) | Dense | 70B | 70B |
| Llama-3.3-70B (Meta AI, 2024) | Dense | 70B | 70B |
| Mistral-7B (Jiang et al., 2023) | Dense | 7B | 7B |
| Mistral-Small-24B (Mistral AI, 2025) | Dense | 24B | 24B |
| Mixtral-8x7B (Jiang et al., 2024) | MoE | 47B | 13B |
| DeepSeek-V3 (DeepSeek-AI, 2024) | MoE | 671B | 37B |
| GPT-4o-mini (OpenAI, 2024) | Prop. | 8B* | 8B* |
| Qwen-2.5-72B (Qwen et al., 2025) | Dense | 72B | 72B |

definition combined with 4 few-shot examples, and (5) Mis-aligned: definitions from other datasets applied to the target dataset (6 variants, detailed in Appendix A). All prompts follow a consistent template where the classification question is followed by the message and an "Answer:" suffix. We additionally run two DSPy (Khattab et al., 2023) automated optimization conditions on all nine models: DSPy Optimized (automatically selected few-shot examples) and DSPy Aligned (selected examples plus aligned definition). For each dataset-model pair, we randomly sampled 1,000 instances. All inference was conducted via API at temperature 0 for reproducibility.

ganized in two tiers. Six models form the *core* of our analysis, spanning dense architectures (7B–70B parameters) and mixture-of-experts architectures (Mixtral-8x7B, DeepSeek-V3) across the Llama and Mistral families. All mixed-effects regressions (RQ2, RQ3) are scoped to these six models to keep statistical inference consistent. We additionally evaluate three *extended* models—GPT-4o-mini, Llama-3.3-70B, and Qwen-2.5-72B—under every prompting condition on every primary dataset. These verify that our headline accuracy, rescue-rate, and DSF findings extend to a broader, more recent slice of proprietary and open-weights systems. For MoE models, we report both total and active parameters per forward pass. Appendix G details which models contribute to each table and figure.

**Datasets.** We focus on toxicity and hate speech detection across five primary datasets covering a range of online communication environments, from formal news comments to informal gaming chats, each with a different operational definition of toxicity (Table 1). These five form the analytic core and appear in every results table in the main paper. We additionally use three supplementary datasets for targeted robustness and generalization checks: *Jigsaw Unintended Bias*, which extends the DSF panel to $N = 54$ model–dataset pairs and serves as a label-bias robustness check for the rescue-rate analysis; and *SemEval-2018 Irony* together with *Subjectivity* which provide non-toxicity replications of RQ1 and RQ2 (Appendix F). Appendix G maps each dataset to the specific analyses it contributes to.

**Experiments.** We evaluate five prompting conditions: (1) Zero-Shot: task name only with no additional context, (2) Aligned Definition: task name plus the dataset's official definition of the target concept, (3) Few-Shot: task name plus 4 random balanced examples, (4) FS+Def: the aligned

**Statistical Analysis** For RQ1, we examined associations between familiarity metrics and annotation accuracy using regression with cluster-robust standard errors (to account for dependence within models) and partial correlations controlling for dataset. For RQ2 and RQ3, we used mixed effects logistic regression on the six open-weights models to account for repeated observations of the same texts across conditions. This handles the binary outcome (correct vs. incorrect) while modeling the non-independence of the same text being evaluated under multiple prompting conditions. We include random intercepts for text and fixed effects for model, prompting condition, domain, and their interactions.

## 4. Results

Our core analysis covers predictions across 9 models, 5 toxicity datasets, and 10 prompting conditions, plus two DSPy automated optimization conditions. Table 3 summarizes model performance and steerability, while Table 5 presents key findings from our mixed effects regression models.

### 4.1. Answering RQ1: Impact of Model Familiarity

We analyzed whether annotation performance correlates more strongly with text memorization or semantic task definition alignment (Definition-Specific Familiarity, DSF). Table 4 reports the statistical association between three memorization metrics (ROUGE-L, BERTScore, and embedding cosine similarity) and DSF against task performance. Unad-

*Table 3.* Model performance (Accuracy % / AUC) across prompting conditions. Misaligned averages 6 definition-swap conditions (full breakdown in Appendix D).

| Model | Zero-Shot | Aligned Def | Few-Shot | FS + Def | DSPy Opt. | DSPy Aligned | Misaligned | Overall |
|---|---|---|---|---|---|---|---|---|
| Llama-3.1-8B | 76.7/.832 | 80.1/.843 | 75.3/.822 | 75.4/.824 | 72.3[†]/.809 | 73.5/.821 | 77.0/.822 | 76.8/.818 |
| Llama-3.1-70B | 79.8/.881 | 82.1/.888 | 81.2/.892 | 82.1/.898 | 79.9 /.**889** | 82.6/.899 | 78.1/.859 | 79.7/.869 |
| Llama-3.3-70B | 80.5/**.885** | 82.3/**.892** | 82.0/.896 | 83.0/.904 | 77.5[†]/.884 | 81.3/**.904** | 79.0/.863 | 80.1/**.874** |
| Mistral-7B | 80.5/.858 | 81.3/.787 | 81.5/.852 | 80.8/.838 | 82.0 /.827 | 81.7/.825 | **81.4**/.823 | 81.2/.819 |
| Mistral-Small-24B | 78.0/.868 | 81.0/.863 | 80.8/.870 | 82.3/.879 | 79.3 /.867 | 81.2/.876 | 80.7/.850 | 80.5/.852 |
| Mixtral-8x7B | 81.0/.868 | 82.7/.868 | 82.4/.868 | 80.9/.859 | 82.8 /.860 | 82.7/.861 | 81.2/.843 | 81.6/.850 |
| DeepSeek-V3 | 81.3/.874 | 83.0/.881 | 82.6/.890 | **83.8**/.893 | 80.9 /.877 | 83.3/.890 | 80.7/.856 | 81.6/.865 |
| GPT-4o-mini | 81.6/.871 | **83.3**/.875 | **84.1**/.880 | 83.3/.880 | **84.4** /.883 | **84.3**/.883 | 81.1/.857 | **82.3**/.862 |
| Qwen-2.5-72B | **83.3**/.876 | 82.2/.879 | 83.8/.894 | 83.2/.892 | 82.5 /.887 | 83.2/.891 | 81.2/**.868** | 82.2/.872 |
| **Cond. Avg** | 80.3/.868 | 82.0/.864 | 81.5/.874 | 81.6/.874 | 80.2 /.865 | 81.5/.872 | 80.3/.849 | 80.8/.853 |

[†]DSPy Opt. degrades Llama-3.1-8B (76.7→72.3) and Llama-3.3-70B (80.5→77.5), consistent with Tan et al. (2025).

justed correlations suggest a strong negative relationship between memorization and accuracy ($r = -0.80$ for ROUGE-L, $-0.76$ for BERTScore, $-0.71$ for embedding similarity). However, these are not directly interpretable because memorization scores are correlated with dataset difficulty (e.g., Jigsaw is both high-memorization and hard), making them sensitive to between-dataset differences.

After controlling for dataset difficulty using partial correlations, all three memorization metrics show no positive association ($r = -0.19, -0.15, -0.16$ respectively). In contrast, consensus DSF is significantly positively associated with performance ($r = +0.41, p = 0.003, N = 54$ model–dataset pairs), an effect that remains robust even after controlling for text complexity (negative log-likelihood). This confirms that DSF captures semantic alignment rather than simple processing fluency, indicating that ***models whose internal concepts align with the task definition perform significantly better***. The same direction is observed across each of the six embedding models used to compute DSF (per-embedding partial $r$ ranges from $+0.30$ to $+0.49$; see Appendix E.1). The partial correlation is further amplified in few-shot settings (partial $r = +0.62, N = 54$); when an aligned definition is provided it attenuates to partial $r = +0.25$ ($N = 54, p = 0.085$), consistent with explicit definitions compressing between-model variance in DSF.

Crucially, ***the relation also holds on two non-safety domains*** (irony and subjectivity detection), where DSF remains positively associated with zero-shot accuracy (partial $r = +0.34, N = 18$ model–dataset pairs); see Appendix F.

### 4.2. Answering RQ2: Impact of Knowledge Injection

Table 5 presents results from mixed effects logistic regression models examining factors associated with annotation correctness and error correction. Odds Ratios (OR) indicate the multiplicative change in odds: $OR = 2.0$ means the odds double when a factor is present, while $OR = 0.5$ means the odds are halved.

*Table 4.* Correlations between familiarity metrics and zero-shot annotation accuracy on the extended $N = 54$ panel. Raw correlations are misleading due to dataset confounds; partial correlations (controlling for dataset) reveal that definition alignment (consensus DSF) is positively associated with performance while three distinct memorization metrics are not.

| Metric | Raw $r$ | Partial $r$ |
|---|---|---|
| Text Memorization (ROUGE-L) | $-0.80$ | $-0.19$ |
| Memorization (BERTScore) | $-0.76$ | $-0.15$ |
| Memorization (Embedding Similarity) | $-0.71$ | $-0.16$ |
| Definition Familiarity (consensus DSF) | $+0.74$ | $+0.41$ |

Domain is most strongly associated with correctness: Gaming is approximately 14 times easier than Forum ($OR = 13.85$), while Social Media is 5 times easier ($OR = 5.45$). Notably, aligned definitions provide only minimal improvement over the zero-shot baseline (+2.2% accuracy on average, Table 3), while misaligned definitions perform worse than aligned (-1.8% and $OR = 1.08$ for aligned vs. misaligned). DSPy automated optimization produces similarly modest gains on average: 80.2% (DSPy Opt.) and 81.5% (DSPy Aligned) are within 0.5pp of the zero-shot (80.3%) and aligned definition (82.0%) baselines respectively, confirming that the performance ceiling reflects model-internalized priors rather than suboptimal prompt design. Gains are heterogeneous across models: GPT-4o-mini improves slightly (84.4%), while both Llama models degrade under DSPy Opt. — Llama-3.1-8B from 76.7% to 72.3% and Llama-3.3-70B from 80.5% to 77.5% — consistent with Tan et al. (2025), who find that DSPy optimizers can degrade performance relative to the unoptimized baseline across Llama-family models, with 10th-percentile relative gains as low as $-13.9\%$.

**Zero-shot Correctness.** Zero-shot correctness is highly associated with prompted correctness ($OR = 6.43$). Conditional on the other covariates in the model, a correct zero-shot response corresponds to $6.4\times$ higher odds of being correct when provided aligned definitions or examples. This

suggests that prompting is more effective at consolidating correct answers than at rescuing errors.

*Table 5.* Key findings from mixed effects logistic regression. OR $> 1$ indicates higher odds of a correct annotation. Reference categories: Forum (domain), Llama-8B (model). Full model specifications in Appendix C. All $p < 0.001$.

| Factor | OR [95% CI] |
| --- | --- |
| *Factors Associated with Correctness:* | |
| Gaming (vs. Forum) | 13.85 [13.31, 14.41] |
| Social Media (vs. Forum) | 5.45 [5.27, 5.63] |
| Aligned Definition | 1.08 [1.04, 1.11] |
| Aligned × News | 1.91 [1.80, 2.02] |
| **Zero-shot Correct** | 6.43 [6.28, 6.58] |
| *Factors Associated with Error Correction:* | |
| ZS Confidence (per SD) | 0.84 [0.82, 0.87] |

*Table 6.* Rescue rate by model, aggregated across prompted conditions. Rescue rate is $P$(correct | prompted, zero-shot wrong).

| Model | Rescue Rate | 95% CI | N (ZS err.) |
| --- | --- | --- | --- |
| Mistral-small-24B | 44.2% | [43.1, 45.2] | 8,536 |
| DeepSeek-V3 | 38.1% | [36.9, 39.2] | 7,264 |
| Llama-3.1-8B | 37.5% | [36.5, 38.5] | 9,040 |
| GPT-4o-mini | 35.7% | [34.6, 36.8] | 7,152 |
| Mixtral-8x7B | 34.6% | [33.5, 35.7] | 7,416 |
| Mistral-7B | 31.6% | [30.5, 32.6] | 7,568 |
| Llama-3.1-70B | 31.3% | [30.3, 32.4] | 7,848 |
| Llama-3.3-70B | 30.4% | [29.4, 31.4] | 7,592 |
| Qwen-2.5-72b | 27.8% | [26.8, 28.9] | 6,504 |

*Rescue Rate* is the probability that prompting corrects a zero-shot error. The overall rescue rate is only 34.8%, meaning nearly two-thirds of errors resist correction. Decision stickiness similarly replicates on the non-safety domains, with an overall few-shot rescue rate of 45.0% across $N = 3,052$ zero-shot errors (Appendix F). Figure 3 shows an inverted-U: rescue probability peaks at 51.8% for moderate confidence (0.6–0.7) and falls sharply to 20.8% for high-confidence errors ($> 0.9$), the most statistically reliable region of the curve. Model 4 corroborates this trend: each standard-deviation increase in zero-shot confidence reduces rescue odds by 16% ($OR = 0.84$, Table 5). ***When models are confidently wrong, prompting rarely helps.***

The drop at the extreme low-confidence end ($n < 1,000$) is a distinct failure mode rather than a counterexample to decision stickiness. These instances are out-of-distribution and context-deficient: their ROUGE-L is far below the rest of the data (0.025 vs. 0.061) and they are dramatically shorter (median 6 vs. 22 words), consisting largely of single-word gaming fragments (e.g., "noob", "wtf", "bot") whose toxicity depends on context that is unavailable to the model.

Importantly, steerability is not a monotonic function of model size: for example, Mistral-Small-24B rescues more zero-shot errors than Llama-3.1-70B in our experiments, despite being substantially smaller (Table 6). To confirm

this finding is not an artifact of label-bias issues known to affect the standard Jigsaw dataset (e.g., systematic flagging of African American Vernacular English as toxic), we replicated the rescue rate analysis on the Jigsaw Unintended Bias dataset (Borkan et al., 2019), which was explicitly designed to reduce identity-group annotation bias. Rescue rates remain uniformly below 50% across all nine models (Table 19), confirming that decision stickiness is not specific to Jigsaw's label properties.

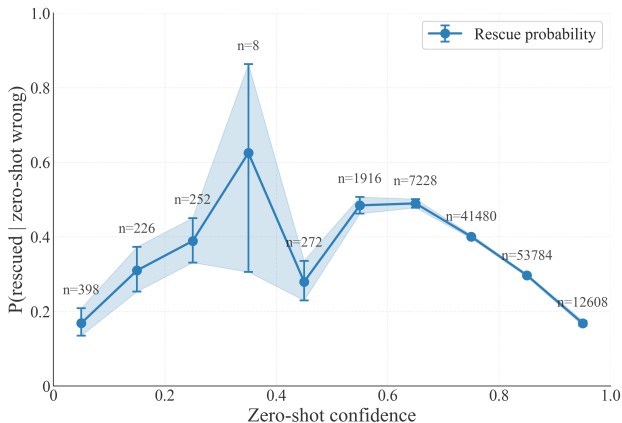

*Figure 3.* Rescue probability vs. zero-shot confidence for zero-shot errors. The inverted-U shape exhibits two distinct failure modes (analyzed in *Confidence and Decision Stickiness*): *decision stickiness* at high confidence (right tail) and an out-of-distribution effect at very low confidence (left tail).

### 4.3. Answering RQ3: Misalignment Impact

We analyze 6 misaligned definition conditions to investigate how LLMs respond to incorrect instructions (Table 7). The results reveal several surprising patterns.

We find that LLMs are responsive to definition scope. Narrow definitions (requiring targeting based on race, religion, gender, etc.) produce prediction bias of $-7\%$ to $-12\%$ (under-prediction), while broad definitions produce bias of $+9\%$ to $+13\%$ (over-prediction). This shift in prediction rates demonstrates that LLMs do not simply ignore misaligned instructions; they adjust decision threshold to match the provided criteria. This responsiveness is a double-edged sword: while it enables definition-based steering, it also means models will confidently apply incorrect definitions.

The worst misaligned condition (gaming toxicity: 76.4%) is 5.6% below aligned, while the best (Fox News hate speech: 82.6%) actually *outperforms* the aligned definition by 0.6%. Knowledge built up during training and tuning constrains how much definitions can shift behavior: even with "wrong" definitions, models remain relatively stable, indicating persistent influence from these internalized task concepts.

Surprisingly, some misaligned definitions provide net benefit over zero-shot (rescuing more errors than they corrupt). For instance, using a "hate speech" definition can improve

performance on general toxicity tasks by focusing the model on clearer criteria, even if technically misaligned. This finding challenges the assumption that the "correct" definition is always optimal - empirically, some misaligned definitions better match the model's internal concept.

*Table 7.* Misalignment analysis: LLMs adjust prediction thresholds based on definition, yet maintain high confidence. Rows are ordered by definition breadth (Narrow → Broad). Confidence remains 84–91% regardless of alignment. All values are percentages.

| Condition | Acc. | Change | Bias | Conf. |
|---|---|---|---|---|
| Aligned definition | 82.0 | 11.3 | ref | 88.1 |
| Fox News hate speech | **82.6** | 17.5 | −7.2 | **90.6** |
| Twitter hate speech | 78.5 | **23.1** | −12.1 | 88.1 |
| GameTox toxicity | 79.6 | 15.4 | +0.5 | 86.5 |
| Olid offensive | 81.9 | 10.9 | +5.1 | 89.1 |
| General toxicity | 81.1 | 9.2 | +8.8 | 85.0 |
| Gaming toxicity | 76.4 | 13.3 | **+13.1** | 88.5 |

**Critical Calibration Failure.** Models *remain confident even when applying misaligned definitions* (Table 7), with confidence levels remaining essentially unchanged from the zero-shot baseline (Table 8). The narrowest definition (Fox News hate speech) produces the *highest* confidence (91.2%), not the lowest. This represents a fundamental calibration failure: models should exhibit increased uncertainty when instructions contradict their training distribution, but they do not. As shown in Figure 4, all conditions (zero-shot, aligned, and misaligned) exhibit similar overconfidence patterns, with calibration curves falling below the diagonal. Critically, there is no meaningful separation between aligned and misaligned definitions, meaning *practitioners cannot rely on confidence scores to detect definition errors*.

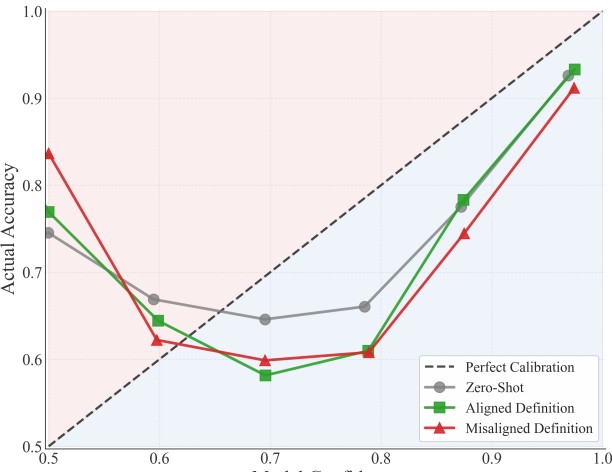

*Figure 4.* Calibration curves showing confidence vs. actual accuracy across conditions. All conditions exhibit overconfidence (below the diagonal), with no meaningful separation between aligned and misaligned definitions: models cannot distinguish when they are applying incorrect instructions.

Across all models and datasets, definition choice produces

*Table 8.* Mean self-reported confidence (%) by prompting condition. Zero-shot confidence is essentially unchanged relative to aligned and misaligned definitions.

| Condition | Mean Confidence |
|---|---|
| Zero-shot | 87.0% |
| Aligned definition | 88.1% |
| Misaligned - Fox News hate speech | 90.6% |
| Misaligned - Olid offensive | 89.1% |
| Misaligned - Gaming toxicity | 88.5% |
| Misaligned - Twitter hate speech | 88.1% |
| Misaligned - GameTox toxicity | 86.5% |
| Misaligned - General toxicity | 85.0% |

17% accuracy variation (e.g., for Fox News, accuracy ranges from 61.6% with the "Gaming Toxicity" definition to 78.4% with the "Twitter Hate Speech" definition), while model choice produces only 5% variation on average. This implies that for annotation tasks, *careful definition design is more impactful than selecting larger or more capable models*.

Models showing high rescue rates under misalignment also show high corruption rates (Table 6 and Table 18). Mistral-Small-24B achieves the highest rescue rate (73.7% with Twitter hate speech) but also the highest corruption rate (21.0%). This creates a fundamental trade-off: *models that are easily steered by definitions can be steered in both beneficial and harmful directions*.

## 5. Discussion

Our results suggest that LLM-based annotation is often constrained less by prompt design than by the fit between a model's internalized concept and the task's definition. Across models and datasets, Definition-Specific Familiarity (DSF) is positively associated with accuracy after accounting for between-dataset differences, whereas three distinct text memorization metrics (ROUGE-L, BERTScore, embedding similarity) all fail to. This pattern supports a concept-level view of annotation: performance depends mainly on whether the model's implicit decision rule matches the dataset's definition (i.e., where it draws the toxic/non-toxic line), rather than on text-level memorization.

**Definition alignment matters more than contamination for annotation.** A common concern is that benchmark performance reflects memorization of test examples. In our setting, cross-dataset association between memorization (whether measured via ROUGE-L, BERTScore, or embedding similarity) and accuracy, once we control for difficulty and domain differences, provides little explanatory power at the model level, while DSF shows a consistent positive correlation. Practically, this shifts the emphasis from auditing overlap with benchmark text toward evaluating whether the model's *concept boundary* matches the intended definition.

**Prompting has limited ability to overturn wrong decisions.** We find substantial *decision stickiness*: most zero-shot errors persist under aligned definitions and few-shot examples, and high-confidence errors are especially resistant to correction (Figure 3, Table 5). This indicates that prompting primarily stabilizes or modestly improves predictions the model already gets right, rather than serving as a reliable mechanism for error recovery. We further test whether iterative, history-aware prompting can break this *stickiness*: a three-turn rescue sequence that progressively adds few-shot examples, the aligned definition, and an explicit reconsideration prompt raises rescue from 7.5% at Turn 1 to only 18.7% at Turn 3, and lifts high-confidence rescue to just 8.5% (Appendix D.1). Decision stickiness therefore persists under iterative correction, not only one-shot prompting. Notably, steerability is not monotonic in model size (Table 6), implying that larger models are not necessarily easier to steer for annotation workloads.

**Misaligned definitions shift behavior without reliably signaling risk.** When given semantically related but mismatched definitions, models change their prediction rates in systematic ways (Table 7, see also Table 16 in Appendix), consistent with literal application of the provided criteria. However, accuracy often degrades only modestly and sometimes even improves relative to the aligned condition. One interpretation is that the model's training supplies a strong prior that anchors behavior, providing a performance floor even under imperfect instructions. Another possibility is that dataset label boundaries may be closer to some alternative formulations than the dataset's stated definition, especially for inherently subjective constructs. Regardless, the key operational point is that *definition wording is a first-order experimental factor*: in our study, definition choice can induce substantial performance variation, so treating the definition as fixed can meaningfully understate uncertainty in downstream analyses.

**Confidence is conditional, not diagnostic of definition validity.** A central deployment risk is that models remain highly confident under misalignment, and calibration curves show little separation between aligned and misaligned conditions (Figure 4). Importantly, this does *not* imply that the model "should know" the definition is wrong; rather, it suggests that the reported confidence reflects certainty *given the provided instructions*, not a meta-uncertainty about whether those instructions match the intended labeling policy. As a result, confidence thresholds are poorly suited for detecting definition errors or definition-dataset mismatch.

**Practical implications for LLM annotation pipelines.** These findings suggest three lightweight safeguards. First, measure *definition alignment* before large-scale annotation (e.g., DSF-style checks) to anticipate which model–definition pairs are likely to work well. Second, *stress-test*

*definitions* by evaluating multiple plausible formulations and reporting sensitivity (e.g., changes in bias, rescue, and corruption), rather than assuming a single "correct" wording is robust. Third, avoid using confidence as a proxy for definition appropriateness.

**Limitations and future work.** Our conclusions are based on toxicity-related binary classification with concept-substitution misalignment; other task types (multi-class, span labeling, or open-ended judging) may exhibit different failure modes. We report DSF as a consensus across six diverse sentence encoders, mitigating embedding-choice sensitivity (Appendix E.1). Robustness to alternative similarity measures (e.g., Mahalanobis distance, learned metrics) and elicitation prompts remains an important next step. Importantly, all evaluated models are instruction-tuned, and we cannot distinguish whether low rescue rates reflect capability limitations or intentional design choices: alignment training may deliberately limit steerability for safety reasons, making some "stickiness" a feature rather than a bug. Our design also cannot isolate which training stage (pre-training, instruction tuning, or RLHF/DPO) produces the observed concept anchoring; we use *model-internalized priors* as a stage-agnostic term, and disentangling the sources would require comparing base and post-trained variants of the same model. Finally, our findings are correlational: establishing that DSF causally drives accuracy (or confidence causally drives stickiness) would require interventions such as targeted fine-tuning that manipulates DSF while holding other factors constant. Future work could test stronger forms of misalignment, alternative multi-turn correction strategies beyond the protocol in Appendix D.1 (e.g., self-consistency, debate, or critique-and-revise), and scalable procedures for selecting model–definition pairs under explicit constraints.

# 6. Conclusion

We study how model-internalized priors interact with prompt-based steering in LLM annotation. Across 9 models and 5 toxicity datasets, we find that performance is best explained by *definition-specific familiarity* (DSF): controlling for dataset difficulty, DSF is positively associated with accuracy (partial $r = +0.41$, $N = 54$), while three distinct memorization metrics (ROUGE-L, BERTScore, embedding similarity) are not. Prompting often fails to correct mistakes, as most zero-shot errors persist even under aligned definitions and few-shot examples. Models can confidently follow *misaligned* definitions, making confidence an unreliable indicator of definition correctness. Finally, definition wording induces larger performance swings than model choice, highlighting definition design as crucial. Practically, LLM annotation pipelines should prioritize measuring definition alignment as an early indicator of performance rather than relying on larger models or confidence-based strategies.

## Acknowledgment

This work is supported by the Caltech Linde Center for Science, Society, and Public Policy (LCSSP). R. Michael Alvarez is Flintridge Foundation Professor of Political and Computational Social Science at Caltech. We thank Andrea Boonyarungsrit, Grant Cahill, Jonathan Lane, Min Kim, and Fereshteh Soltani for prior collaboration that helped inform some of the motivating challenges behind this work. The views and opinions expressed in this paper are solely those of the authors and do not necessarily reflect those of the individuals acknowledged or their affiliated organizations.

## Impact Statement

This work examines how model-internalized priors interact with prompt-based steering in LLM annotation tasks. Our findings have implications for practitioners deploying LLMs as annotators or judges. Positively, identifying failure modes such as decision stickiness and miscalibration under definition misalignment can help practitioners develop better validation procedures and avoid over-reliance on confidence scores. However, our results also reveal that models will confidently apply incorrect definitions when provided, which could lead to systematic errors if task specifications are poorly designed.

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

## A. Definition Texts Used

This section provides the exact definition texts given to models, organized by scope (narrow to broad). Understanding definition scope is critical for interpreting misalignment results.

### A.1. Narrow Scope Definitions

These definitions require identity-based targeting; general insults without identity targeting are explicitly excluded.

**Twitter Hate Speech (Davidson et al., 2017).**

> **Hate Speech:** Language that expresses hatred towards a targeted group or is intended to be derogatory, to humiliate, or to insult the members of the group based on attributes such as race, religion, ethnic origin, sexual orientation, disability, or gender.
>
> **Not Hate Speech:** Content that does not express hatred or target groups based on identity attributes. Note: Offensive language that contains profanity or insults but does not target identity groups is NOT considered hate speech.

**Fox News Hate Speech.**

> **Hateful:** Language that explicitly or implicitly threatens or demeans a person or a group based upon a facet of their identity such as gender, ethnicity, or sexual orientation. This includes comments that express prejudice, discrimination, or hostility toward individuals or groups based on their protected characteristics.
>
> **Non-hateful:** Comments that do not contain language that threatens or demeans individuals or groups based on their identity characteristics.

### A.2. Medium Scope Definitions

These definitions include insults and threats regardless of identity targeting, but still require explicit negative content.

**GameTox Multi-Category Definition.**

> **Toxic:** Any message that falls into one or more of these categories: (1) Hate and Harassment: Utterances with identity-based hate or harassment; (2) Threats: Utterances with threats of violence, physical harm, terrorism, or doxing; (3) Extremism: Utterances with extremist views or recruitment attempts; (4) Insults and Flaming: Insults or attacks on another player (not identity-based); (5) Other Offensive Texts: Any message that harms reasonable enjoyment of the game.
>
> **Not toxic:** Utterances without any toxicity from the above categories.

**OLID Offensive Language (Zampieri et al., 2019).**

> **Offensive:** This post contains offensive language or a targeted (veiled or direct) offense. This includes any form of non-acceptable language (profanity), insults, threats, and posts containing profane language or swear words.
>
> **Not Offensive:** This post does not contain offense or profanity. The content may express opinions or criticism but does not include any form of offensive language, insults, or threats.

### A.3. Broad Scope Definitions

These definitions capture any content that is "rude," "disrespectful," or "discouraging" - very broad criteria.

**Gaming Toxicity (CONDA) (Weld et al., 2021).**

**Toxic:** Utterances containing explicit or implicit toxicity in gaming context. *Explicit Toxicity:* Direct toxic expressions such as profanity, personal attacks, or insults directed at other players. *Implicit Toxicity:* Subtler forms including sarcasm, veiled insults, or coded language that requires conversational context to identify as harmful.

**Not Toxic:** Utterances related to gameplay actions, neutral communication, or content that does not discourage player engagement. General expressions of frustration not directed at specific players (e.g., "damn!" or "ugh") are not toxic.

### General Toxicity (Perspective API/Jigsaw) (cjadams et al., 2018).

A rude, disrespectful, or unreasonable comment that is likely to make someone leave a discussion.

*Table 9.* Definition scope classification. **Narrow** definitions require identity-based targeting (e.g., race, religion, gender) to qualify as toxic, excluding general insults. **Medium** definitions include direct insults, threats, and profanity regardless of identity targeting. **Broad** definitions capture any content that is rude, disrespectful, or discouraging, including implicit and contextual toxicity. When applied to general toxicity tasks, narrow definitions produce under-prediction while broad definitions produce over-prediction.

| Definition | Scope |
|---|---|
| Twitter Hate Speech | Narrow |
| Fox News Hate Speech | Narrow |
| GameTox Toxicity | Medium |
| OLID Offensive | Medium |
| Gaming Toxicity (CONDA) | Broad |
| General Toxicity | Broad |

### A.4. Non-Safety Task Definitions

The following definitions were used in the generalization experiments reported in Appendix F. The irony and subjectivity definitions were used as the "aligned definition" prompt in the annotation experiments (Tables 25 and 26); the five stance definitions were used as the elicitation context for DSF on stance detection. All definitions operationalize the target concept following the framing of the original task papers and do not fit the narrow/medium/broad scope schema of Table 9, which applies only to the toxicity datasets.

### Irony Detection (SemEval-2018 Task 3) (Van Hee et al., 2018).

**Ironic:** A statement where the intended meaning is opposite to or incongruent with the literal meaning, often used to express humor, criticism, or sarcasm. Irony involves a discrepancy between what is said and what is meant, or between expectations and reality.

**Not Ironic:** A statement where the intended meaning matches the literal meaning, with no incongruence between expression and intent.

### Subjectivity Detection (Pang & Lee, 2004).

**Subjective:** A sentence that expresses personal opinions, evaluations, emotions, or speculations. Subjective sentences reflect the author's point of view, feelings, or interpretations rather than objective facts.

**Objective:** A sentence that presents factual information that can be verified independently of the author's personal opinion or emotional state.

**Stance Detection (SemEval-2016 Task 6) (Mohammad et al., 2016).** The stance task uses five targets, each with its own definition. Unlike the other tasks in our study, stance is framed as a choice between two explicit alternatives (*in favor / against*) rather than as classification against a single concept definition, as discussed in Appendix F.

**Stance toward Legalization of Abortion.** *In favor:* The tweet expresses support for or agreement with the legalization of abortion, including women's right to choose. *Against:* The tweet expresses opposition to the legalization of abortion, including pro-life positions or moral objections.

**Stance toward Atheism.** *In favor:* The tweet expresses support for, agreement with, or positive views toward atheism or non-belief in God. *Against:* The tweet expresses opposition to, criticism of, or negative views toward atheism.

**Stance toward Climate Change.** *In favor:* The tweet expresses belief in or concern about climate change as a real and serious issue requiring action. *Against:* The tweet expresses skepticism about, denial of, or opposition to claims about climate change.

**Stance toward Feminist Movement.** *In favor:* The tweet expresses support for or agreement with the feminist movement and gender equality. *Against:* The tweet expresses opposition to, criticism of, or negative views toward the feminist movement.

**Stance toward Hillary Clinton.** *In favor:* The tweet expresses support for, agreement with, or positive views toward Hillary Clinton. *Against:* The tweet expresses opposition to, criticism of, or negative views toward Hillary Clinton.

## B. Prompts

All prompts follow a consistent template structure. The base classification prompt presents the task, followed by the text to classify, and ends with an "Answer:" suffix. The confidence elicitation suffix (Figure 2) is appended to all classification prompts.

**Experimental Conditions.** Table 10 summarizes the 10 manually-specified prompting conditions together with the 2 DSPy automated optimization conditions used in our experiments.

*Table 10.* Description of experimental conditions.

| Condition | Description | Aligned |
|---|---|---|
| zero_shot | No definition provided | – |
| aligned_definition | Definition matches dataset | Yes |
| few_shot | 4 examples, no definition | – |
| few_shot_aligned_def | 4 examples + aligned definition | Yes |
| misaligned_gaming_toxicity | Gaming toxicity definition | No |
| misaligned_general_toxicity | Perspective API definition | No |
| misaligned_gametox_toxicity | GameTox definition | No |
| misaligned_foxnews_hate_speech | Fox News hate speech definition | No |
| misaligned_twitter_hate_speech | Twitter hate speech definition | No |
| misaligned_olid_offensive | OLID offensive definition | No |
| dspy_optimized | DSPy-selected few-shot examples | – |
| dspy_aligned | DSPy-selected few-shot examples + aligned definition | Yes |

### B.1. DSPy Automated Optimization

We additionally evaluate two automated prompt optimization conditions using DSPy (Khattab et al., 2023), which selects few-shot demonstrations from a labeled validation pool (of 100 examples) to maximize accuracy. **DSPy Optimized** uses a bare classification signature with no human-written definition (automated counterpart to few_shot). **DSPy Aligned** additionally includes the dataset's aligned definition in the instruction field (automated counterpart to few_shot_aligned_def). Optimization is run independently per model–dataset pair.

### B.2. Continuation Prompt

To measure text-level familiarity (ROUGE-L), we split each text instance at the 40% mark by word count. We prompt the model with the first 40% (prefix) using the following template:

Continue the following passage faithfully: [PREFIX]

The model generates a continuation using greedy decoding (temperature 0.0). We then compute the ROUGE-L F1 score between the generated continuation and the remaining 60% of the ground truth text (suffix).

**Additional memorization metrics.** To confirm that ROUGE-L's limitations as a memorization proxy do not drive our findings, we compute two semantically rich alternatives over the same continuation/suffix pairs. Both use the `all-MiniLM-L6-v2` encoder—the same encoder as DSF—so any sign difference between memorization and DSF reflects signal type rather than encoder choice. **Embedding Similarity** is the cosine similarity between the sentence embeddings of the generated continuation and the ground-truth suffix. **BERTScore** (Zhang et al., 2020) computes token-level F1 via pairwise cosine similarities of contextual token embeddings of the two texts. Both metrics are averaged at the model–dataset level before correlation analysis.

## C. Mixed Effects Regression Models

We fit four mixed effects logistic regression models to analyze annotation correctness and error correction. All models include message-level random intercepts to account for repeated measures across experimental conditions. Models were estimated using Bayesian variational inference (BinomialBayesMixedGLM in statsmodels).

**Model 1: Domain-Alignment Effects**   **Research Question:** What factors predict correct annotation?

**Formula:**        `correct ~ C(model) + C(condition) + C(domain) + is_aligned_defn +`
`C(domain):is_aligned_defn + (1|message)`

**Reference categories:** Llama-3.1-8B (model), Forum (domain), Misaligned definition (condition)

*Table 11.* Model 1: Full results for domain-alignment effects.

| Predictor | Coefficient | OR [95% CI] |
|---|---|---|
| Intercept | 1.549 | 4.71 [4.63, 4.78] |
| *Model Effects (vs. Llama-8B):* | | |
| DeepSeek-V3 | 0.499 | 1.65 [1.58, 1.72] |
| Llama-3.1-70B | 0.237 | 1.27 [1.22, 1.32] |
| Mistral-7B | 0.384 | 1.47 [1.42, 1.52] |
| Mistral-Small-24B | 0.326 | 1.39 [1.33, 1.44] |
| Mixtral-8x7B | 0.456 | 1.58 [1.52, 1.64] |
| *Domain Effects (vs. Forum):* | | |
| Gaming | 2.628 | 13.85 [13.31, 14.41] |
| News | −0.048 | 0.95 [0.92, 0.99] |
| Social Media | 1.695 | 5.45 [5.27, 5.63] |
| *Alignment Effects:* | | |
| is_aligned_defn | 0.073 | 1.08 [1.04, 1.11] |
| Gaming × Aligned | 0.419 | 1.52 [1.40, 1.65] |
| News × Aligned | 0.645 | 1.91 [1.80, 2.02] |
| Social Media × Aligned | 0.333 | 1.40 [1.32, 1.47] |
| Random Effects SD | 1.156 | 3.18 [3.11, 3.24] |

**Model 2: Model-Alignment Interaction**   **Research Question:** Do different models benefit differently from aligned definitions?

**Formula:**        `correct ~ C(model) + C(condition) + C(domain) + is_aligned_defn +`
`C(model):is_aligned_defn + (1|message)`

*Table 12.* Model 2: Model $\times$ Alignment interactions.

| Model | Alignment Interaction | OR [95% CI] |
|---|---|---|
| Llama-3.1-8B (ref) | 0.195 | 1.22 [1.18, 1.25] |
| Llama-3.1-70B | 0.437 | 1.55 [1.44, 1.67] |
| DeepSeek-V3 | 0.320 | 1.38 [1.28, 1.48] |
| Mistral-Small-24B | 0.202 | 1.22 [1.14, 1.32] |
| Mixtral-8x7B | 0.100 | 1.11 [1.03, 1.19] |
| Mistral-7B | 0.078 | 1.08 [1.01, 1.16] |

Llama-3.1-70B benefits most from alignment (OR=1.55), despite having the lowest rescue rate. This suggests larger models have stronger priors that are harder to override, but can better utilize guidance when provided.

**Model 3: Zero-Shot Predictor**  **Research Question:** Does zero-shot correctness predict prompted performance?

**Formula:**       `correct ~ C(model) + C(condition) + C(domain) + zero_shot_correct + (1|message)`

*Table 13.* Model 3: Zero-shot as predictor of prompted correctness.

| Predictor | Coefficient | OR [95% CI] |
|---|---|---|
| Zero-shot Correct | 1.861 | 6.43 [6.28, 6.58] |
| *Domain Effects (vs. Forum):* | | |
|    Gaming | 2.124 | 8.36 [8.01, 8.73] |
|    News | 0.177 | 1.19 [1.15, 1.24] |
|    Social Media | 1.487 | 4.43 [4.26, 4.60] |

**Model 4: Rescue Prediction**  **Research Question:** Among zero-shot errors, what predicts whether prompting will correct the error?

**Formula:** `rescued ~ C(model) + C(condition) + C(domain) + zs_conf_std + (1|message)`

This model is fit only on cases where the model was wrong in zero-shot (N=47,672 errors).

*Table 14.* Model 4: Predicting error correction (rescue) among zero-shot errors.

| Predictor | Coefficient | OR [95% CI] |
|---|---|---|
| ZS Confidence (standardized) | $-0.171$ | 0.84 [0.82, 0.87] |
| *Model Effects (vs. Llama-8B):* | | |
|    DeepSeek-V3 | 0.634 | 1.89 [1.75, 2.03] |
|    Llama-3.1-70B | 0.111 | 1.12 [1.04, 1.20] |
|    Mistral-7B | $-0.071$ | 0.93 [0.87, 1.00] |
|    Mistral-Small-24B | 0.514 | 1.67 [1.57, 1.79] |
|    Mixtral-8x7B | 0.296 | 1.35 [1.25, 1.45] |
| *Domain Effects (vs. Forum):* | | |
|    Gaming | 0.982 | 2.67 [2.47, 2.89] |
|    News | 0.580 | 1.79 [1.68, 1.89] |
|    Social Media | 1.331 | 3.78 [3.56, 4.02] |

# D. Additional Experimental Results

*Table 15.* Detailed breakdown of model accuracy (%) under specific misaligned definitions. These values are averaged into the single "Misaligned" column in Table 3. Columns are ordered by definition scope (Narrow → Broad). Models below the horizontal rule are not included in regression analyses; Cond. AUC is for the six open-weights models.

| Model | Misaligned Definitions | | | | | |
| | Twitter Hate | FoxNews Hate | OLID Off. | GameTox Tox. | General Tox. | Gaming Tox. |
|---|---|---|---|---|---|---|
| DeepSeek-V3 | 79.2 | 82.0 | 82.6 | 80.8 | 82.0 | 77.4 |
| Mixtral-8x7B | 79.0 | 83.3 | 82.1 | 80.9 | 83.3 | 78.7 |
| Mistral-7B | 79.5 | 83.8 | 84.0 | 80.3 | 84.5 | 76.5 |
| Mistral-Small-24B | 77.2 | 83.1 | 82.9 | 80.3 | 83.5 | 77.3 |
| Llama-3.1-70B | 77.5 | 81.4 | 80.0 | 78.3 | 78.1 | 73.0 |
| Llama-3.1-8B | 77.3 | 79.8 | 78.2 | 76.8 | 76.9 | 72.7 |
| GPT-4o-mini | 79.8 | 82.3 | 82.1 | 82.0 | 81.6 | 78.5 |
| Llama-3.3-70B | 78.6 | 83.0 | 81.5 | 77.8 | 78.9 | 74.4 |
| Qwen-2.5-72B | 78.6 | 84.3 | 83.9 | 79.7 | 81.0 | 79.3 |
| **Cond. Avg.** | 78.5 | 82.6 | 81.9 | 79.7 | 81.1 | 76.4 |
| **Cond. AUC** | .790 | .850 | .867 | .817 | .857 | .841 |

*Table 16.* Accuracy (%) by Dataset and Misaligned Condition. Bold = best for that dataset. "N/A" indicates the aligned (non-misaligned) condition.

| Dataset | foxnews | gametox | gaming | general | olid | twitter |
|---|---|---|---|---|---|---|
| FoxNews | N/A | 75.6 | 61.6 | 70.6 | 73.8 | **78.4** |
| GameTox | 87.2 | N/A | 90.6 | 90.6 | **90.9** | 85.0 |
| Jigsaw | 75.5 | 75.8 | 70.4 | N/A | **77.4** | 73.0 |
| OLID | 75.8 | 79.7 | 79.7 | **79.9** | N/A | 74.3 |
| Twitter | 88.7 | **89.1** | 80.8 | 85.7 | 84.4 | N/A |

*Table 17.* Model sensitivity: best accuracy (across all conditions) minus worst accuracy under misaligned definitions. Models below the horizontal rule are not included in regression analyses.

| Model | Best Acc. | Worst Acc. | Sensitivity |
|---|---|---|---|
| Llama-3.1-70B | 82.1 | 73.0 | 9.1 (highest) |
| Mistral-7B | 84.5 | 76.5 | 8.0 |
| Llama-3.1-8B | 80.1 | 72.7 | 7.4 |
| DeepSeek-V3 | 83.8 | 77.4 | 6.4 |
| Mistral-Small-24B | 83.5 | 77.2 | 6.3 |
| Mixtral-8x7B | 83.3 | 78.7 | 4.6 (lowest) |
| Llama-3.3-70B | 83.0 | 74.4 | 8.6 |
| GPT-4o-mini | 84.1 | 78.5 | 5.6 |
| Qwen-2.5-72B | 84.3 | 78.6 | 5.7 |

*Table 18.* Corruption rate (%) by model and misaligned condition. High corruption indicates that correct predictions are frequently flipped to incorrect ones when the definition changes. Models below the horizontal rule are not included in regression analyses.

| Model | Twitter | FoxNews | OLID | GameTox | General | Gaming |
|---|---|---|---|---|---|---|
| DeepSeek-V3 | 14.7 | 10.7 | 6.4 | 7.5 | 7.4 | 12.1 |
| Llama-3.1-70B | 16.8 | 12.3 | 6.0 | 7.5 | 8.3 | 13.3 |
| Llama-3.1-8B | 14.0 | 10.9 | 11.1 | 10.4 | 7.6 | 12.4 |
| Mistral-7B | 8.4 | 4.6 | 4.0 | 8.2 | 2.2 | 9.9 |
| Mistral-Small-24B | 21.0 | 11.3 | 5.0 | 8.2 | 3.1 | 10.0 |
| Mixtral-8x7B | 9.0 | 6.5 | 6.7 | 5.6 | 4.8 | 10.4 |
| GPT-4o-mini | 13.2 | 10.3 | 5.7 | 10.9 | 5.7 | 9.0 |
| Llama-3.3-70B | 15.2 | 10.0 | 5.4 | 6.9 | 8.3 | 12.0 |
| Qwen-2.5-72B | 12.6 | 7.2 | 4.6 | 7.4 | 8.6 | 9.6 |

*Table 19.* Rescue rate by model on the Jigsaw Unintended Bias dataset (Borkan et al., 2019), which was designed to reduce identity-group annotation bias. Rescue rate is $P(\text{correct} \mid \text{prompted, zero-shot wrong})$, computed over all non-zero-shot conditions (aligned definition, few-shot, few-shot+definition, DSPy Optimized, DSPy Aligned, and five misaligned conditions), matching the methodology of Table 6. All models remain well below 50%, confirming that decision stickiness is not an artifact of Jigsaw's known label-bias limitations.

| Model | Rescue Rate | 95% CI | N (ZS err. obs.) |
|---|---|---|---|
| DeepSeek-V3 | 38.2% | [36.4, 40.1] | 2,640 |
| Mistral-Small-24B | 35.6% | [33.9, 37.4] | 2,810 |
| Llama-3.1-8B | 34.3% | [32.5, 36.0] | 2,790 |
| GPT-4o-mini | 31.4% | [29.6, 33.3] | 2,350 |
| Mixtral-8x7B | 31.0% | [29.3, 32.8] | 2,790 |
| Llama-3.1-70B | 30.8% | [29.1, 32.5] | 2,870 |
| Llama-3.3-70B | 28.2% | [26.6, 29.9] | 2,790 |
| Mistral-7B | 27.8% | [26.1, 29.5] | 2,680 |
| Qwen-2.5-72B | 24.9% | [23.2, 26.7] | 2,330 |

### D.1. Multi-Turn Rescue Experiment

A natural follow-up to the one-shot rescue analysis (Sec. 4.2) is whether iterative, history-aware prompting can break decision stickiness. For each model–dataset pair we sampled 100 zero-shot errors and applied a 3-step rescue sequence: Turn 1 adds a few labeled examples, Turn 2 adds the aligned task definition, and Turn 3 asks the model to reconsider. At each turn the model is shown its previous answers and confidence scores, so later prompts see the full history.

Table 20 reports cumulative rescue rates by model, both overall and restricted to high-confidence errors. Iterative prompting helps modestly — particularly when the aligned definition is added at Turn 2 — but rescue plateaus well below the one-shot rate of 34.8% reported in Table 6, and high-confidence errors remain almost entirely uncorrected (8.5% rescued after three turns).

*Table 20.* Cumulative rescue rates over three multi-turn correction attempts. "High-conf" columns restrict to zero-shot errors with confidence $> 0.9$.

| Model | Rescue@1 | Rescue@2 | Rescue@3 | High-conf.@1 | High-conf.@2 | High-conf.@3 |
|---|---|---|---|---|---|---|
| DeepSeek-V3 | 8.2 | 20.8 | 27.4 | 0.0 | 5.1 | 6.8 |
| Llama-3.1-70B | 10.6 | 14.6 | 19.0 | 2.9 | 5.0 | 8.0 |
| Llama-3.1-8B[†] | 21.7 | 35.8 | 35.8 | 0.0 | 16.7 | 16.7 |
| Mistral-7B | 3.0 | 12.3 | 12.7 | 1.2 | 8.5 | 9.1 |
| Mistral-Small-24B | 7.0 | 12.4 | 12.9 | 4.1 | 9.6 | 9.6 |
| Mixtral-8x7B | 5.8 | 17.1 | 18.5 | 1.9 | 6.5 | 7.4 |
| **Pooled** | 7.5 | 16.3 | 18.7 | 2.0 | 7.1 | 8.5 |

[†]78.8% of sampled items dropped due to classification refusals.

# E. Familiarity Analysis Details

## E.1. DSF Robustness Across Embedding Models

The DSF score is a cosine similarity in an embedding space, so its value depends on the encoder used. To rule out that our results are an artifact of choosing any single embedding model, we recompute DSF using six diverse sentence-embedding models spanning small (MiniLM, mpnet) to large (bge-large, e5-large, instructor-large) open-source encoders, plus a proprietary OpenAI model (text-embedding-3-small). The full model identifiers are:

- MiniLM: `sentence-transformers/all-MiniLM-L6-v2`
- MPNet: `sentence-transformers/all-mpnet-base-v2`
- BGE-large: `BAAI/bge-large-en-v1.5`
- E5-large: `intfloat/e5-large-v2`
- Instructor-large: `hkunlp/instructor-large`
- OpenAI: `text-embedding-3-small`

We then define *consensus DSF* for each model–dataset cell as the unweighted mean DSF across the six encoders, and adopt it as the primary DSF metric in the main text.

The DSF analysis covers all nine evaluated models across six datasets (the five primary datasets plus Jigsaw Unintended Bias (Borkan et al., 2019)), yielding $N = 54$ model–dataset pairs. Model performance is reported in Table 3.

**Per-embedding partial correlations.** Table 21 reports the partial correlation between DSF and zero-shot accuracy (controlling for dataset) separately for each embedding choice. All six embeddings yield a positive partial correlation, ranging from $+0.30$ to $+0.49$. Larger encoders (bge-large, e5-large, OpenAI) tend to yield slightly stronger correlations than the compact MiniLM. The consensus row ($+0.41$) sits at the upper end of this range, indicating that averaging across encoders denoises rather than dilutes the signal.

*Table 21.* DSF–accuracy correlations across six sentence-embedding models on the extended $N = 54$ panel (zero-shot accuracy, partial correlation controls for dataset). Consensus DSF is the per-cell mean across embeddings, and is the metric reported in the main text.

| Embedding model | Raw $r$ | Partial $r$ |
|---|---|---|
| all-MiniLM-L6-v2 | $+0.73$ | $+0.36$ |
| all-mpnet-base-v2 | $+0.77$ | $+0.33$ |
| BAAI/bge-large-en-v1.5 | $+0.74$ | $+0.49$ |
| intfloat/e5-large-v2 | $+0.60$ | $+0.43$ |
| hkunlp/instructor-large | $+0.72$ | $+0.30$ |
| OpenAI text-embedding-3-small | $+0.72$ | $+0.43$ |
| **Consensus (mean across 6)** | $+\mathbf{0.74}$ | $+\mathbf{0.41}$ |

**Pairwise embedding agreement.** DSF values are highly concordant across embedding choices. Table 22 reports the Pearson correlation between DSF score vectors produced by different embedding models on the $N = 54$ panel; all pairwise correlations exceed $0.87$, and most exceed $0.95$. This indicates that the variation in Table 21 reflects what each embedding emphasizes (e.g., lexical vs. semantic, general vs. domain-tuned) rather than disagreement about which model–dataset pairs are high- vs. low-DSF.

*Table 22.* Pairwise Pearson correlations between DSF score vectors across six embedding models, on the $N = 54$ panel. All embeddings agree closely on the relative ordering of model–dataset cells.

| | MiniLM | mpnet | bge | e5 | instr. | OpenAI |
|---|---|---|---|---|---|---|
| MiniLM | 1.00 | 0.97 | 0.98 | 0.93 | 0.97 | 0.97 |
| mpnet | | 1.00 | 0.96 | 0.88 | 0.97 | 0.97 |
| bge-large | | | 1.00 | 0.95 | 0.98 | 0.99 |
| e5-large | | | | 1.00 | 0.94 | 0.93 |
| instructor | | | | | 1.00 | 0.98 |
| OpenAI | | | | | | 1.00 |

**Takeaway.** The DSF–accuracy association is not an artifact of a particular encoder. Across six embedding models the partial correlation remains positive and meaningful, and the embeddings closely agree on which model–dataset pairs score high or low. We therefore report consensus DSF in the main text as a stable, encoder-agnostic operationalization of the metric.

### E.2. Min-K% Prob as a Memorization Baseline

Min-K% Prob (Shi et al., 2024) computes the average log probability of the $k\%$ least likely tokens in a text. Lower (more negative) values indicate the model finds certain tokens surprising, suggesting the text was *not* memorized during training. Computing Min-K% requires per-token log probabilities, which are unavailable for most frontier API models in our evaluation. We therefore use ROUGE-L as the primary memorization baseline, as it is computable via text generation for any model.

To verify that this choice does not change our qualitative conclusions, we applied Min-K% ($k = 20\%$) to the two open-weights models in our panel for which log probabilities are available (Llama-3.1-8B and Mistral-7B, base versions) across the five paper datasets ($N = 10$ model–dataset pairs). Table 23 reports mixed-effects model coefficients (Accuracy $\sim$ metric + (1 | dataset) + (1 | model)) for ROUGE-L, Min-K%, and DSF on this subset. Min-K% and ROUGE-L are directionally consistent—both show negative standardized coefficients while DSF remains positive—confirming that our conclusions do not depend on the choice of memorization metric.

*Table 23.* Mixed-effects model coefficients for memorization metrics and DSF on the $N = 10$ subset (Llama-3.1-8B and Mistral-7B $\times$ 5 paper datasets) where Min-K% Prob is computable. Standardized coefficients allow cross-metric comparison of effect size.

| Metric | Std. Coef. | Raw Coef. | $p$ |
|---|---|---|---|
| Text Memorization (ROUGE-L) | $-0.755$ | $-2.479$ | $0.001$ |
| Memorization (Min-K% Prob) | $-0.596$ | $-0.044$ | $0.036$ |
| Definition Familiarity (DSF) | $+0.838$ | $+0.425$ | $< 0.001$ |

### E.3. Semantic Memorization Metrics: BERTScore and Embedding Similarity

ROUGE-L captures lexical overlap via the longest common subsequence and may therefore underestimate memorization when a model reproduces the *content* of a passage with paraphrased surface form. We additionally evaluate two semantically richer alternatives that operate on the same generated continuations as ROUGE-L, so the comparison isolates the scoring function rather than the generation procedure.

**BERTScore.** BERTScore (Zhang et al., 2020) replaces $n$-gram matching with contextual token-embedding similarity. Each token in the generated continuation $\hat{y}$ and the reference suffix $y$ is encoded with a pretrained contextual encoder, and tokens are greedily aligned by maximum cosine similarity:

$$R_{\text{BERT}} = \frac{1}{|y|} \sum_{y_i \in y} \max_{\hat{y}_j \in \hat{y}} \mathbf{y}_i^\top \hat{\mathbf{y}}_j, \quad P_{\text{BERT}} = \frac{1}{|\hat{y}|} \sum_{\hat{y}_j \in \hat{y}} \max_{y_i \in y} \mathbf{y}_i^\top \hat{\mathbf{y}}_j,$$

with $F_{1,\text{BERT}}$ as their harmonic mean. Because the embeddings are contextual, BERTScore credits paraphrases and near-synonyms that ROUGE-L would penalize. We use `microsoft/deberta-xlarge-mnli` as the encoder (the default recommended by Zhang et al. (2020) for English) and report $F_{1,\text{BERT}}$ as the primary score.

**Embedding cosine similarity.** Embedding similarity moves one level higher, comparing whole-sentence representations rather than token alignments:

$$\text{EmbSim}(\hat{y}, y) = \frac{\phi(\hat{y})^\top \phi(y)}{\|\phi(\hat{y})\| \, \|\phi(y)\|},$$

where $\phi(\cdot)$ is a sentence encoder. This is the loosest of the three notions of textual overlap: a continuation that captures the gist of the reference but rewords it freely can still score high. To ensure that any difference between EmbSim and DSF reflects *what* is being measured (continuation–reference alignment vs. definition–self-description alignment) rather than *which encoder* does the measuring, we use the same `all-MiniLM-L6-v2` encoder as in DSF.

**Spectrum of memorization sensitivity.** Taken together, ROUGE-L, BERTScore, and EmbSim form a progression from purely lexical to fully semantic notions of reproduction. If text memorization were driving annotation accuracy and ROUGE-L were simply too crude to detect it, we would expect the correlation with accuracy to strengthen (or at least become positive) as we relax the lexical constraint. Table 4 shows the opposite: all three text-memorization metrics yield negative partial correlations with accuracy after controlling for dataset, while DSF remains positive. The negative direction is preserved across the full lexical–semantic spectrum, indicating that the issue is not ROUGE-L's lexical sensitivity but that text reproduction itself, however measured, does not predict annotation performance.

### E.4. Definition-Specific Familiarity (DSF) Variants

In the main text, we define **Definition-Specific Familiarity (DSF)** as the cosine similarity between the embedding of the dataset's ground-truth definition and the model's own description of the concept. Below we detail this primary metric alongside two alternative variants we explored.

**DSF (Primary Metric).** This is the metric reported in the main results. We prompt the model with a neutral instruction to describe the concept: *"You are documenting your internal understanding of a labeling concept. Concept: [Concept Name]. Describe, in your own words, what content qualifies as this concept."* We then compute the cosine similarity between the embedding of this generated response and the embedding of the dataset's full definition. We found this approach correlated most strongly with model performance.

**Alternative Variant: Positive DSF.** This variant isolates the "positive" class definition (e.g., what *is* toxicity) from the "negative" class. It uses **multi-prompt elicitation** to reduce variance, averaging similarities across three prompt styles: *Direct* ("You are documenting..."), *Expert* ("As an expert annotator..."), and *Examples* ("Without examples, describe...").

**Alternative Variant: Enhanced DSF.** This metric extends Positive DSF by adding **domain contextualization** (e.g., *"in online gaming chat"* for GameTox) and **negative class alignment** (comparing the model's description of what the concept is *not* to the dataset's negative definition). The final score is the average of the positive and negative alignment scores.

*Table 24.* Familiarity metrics by dataset (averaged across the six open-weights models, zero-shot). DSF is reported as consensus DSF across six sentence encoders. Jigsaw has the lowest DSF and is the hardest dataset; GameTox has the highest accuracy with negligible text memorization, illustrating that conceptual alignment, not memorization, tracks performance.

| Dataset | Domain | ROUGE-L | Consensus DSF | Accuracy |
|---|---|---|---|---|
| Twitter | Social Media | 0.045 | 0.804 | 0.901 |
| GameTox | Gaming | 0.001 | 0.784 | 0.902 |
| OLID | Social Media | 0.063 | 0.710 | 0.751 |
| FoxNews | News | 0.061 | 0.652 | 0.723 |
| Jigsaw | Forum | 0.084 | 0.458 | 0.705 |

## F. Generalization Beyond Toxicity

To test whether our two main findings—the positive DSF–accuracy association (RQ1) and decision stickiness (RQ2)—generalize beyond safety domains, we replicate the core analyses on two non-safety binary classification tasks: *irony detection* (SemEval-2018 Task 3; Van Hee et al. 2018) and *subjectivity detection* (Pang & Lee, 2004). Both tasks use explicit human-authored definitions and binary labels, directly analogous to our toxicity setup. We evaluate the same 9 models from Table 2, sampling 1,000 instances per dataset–model pair as in the main experiments.

**DSF replicates on non-safety tasks.** DSF shows a consistent positive association with zero-shot accuracy across both non-safety datasets (partial $r = +0.343$, $N = 18$ model–dataset pairs), directionally replicating our toxicity finding (partial $r = +0.34$ on the original five datasets, see Section 4.1). The effect is therefore not safety-specific.

**Decision stickiness replicates on non-safety tasks.** The overall one-shot few-shot rescue rate is $45.0\%$ (95% CI $[43.2\%, 46.8\%]$) across 3,052 zero-shot errors, comparable in magnitude to the $34.8\%$ reported for toxicity (Table 6). More than half of zero-shot errors remain uncorrected on these non-safety tasks as well. Table 25 summarizes accuracy by condition (averaged across the two datasets); Table 26 reports the per-model rescue rates.

*Table 25.* Accuracy (%) by condition on the two non-safety datasets (irony and subjectivity), averaged across the two datasets. Sampling and inference settings match the main experiments.

| Model | Zero-Shot | Aligned Def | Few-Shot |
|---|---|---|---|
| Llama-3.3-70B | 73.3 | 70.4 | 86.2 |
| Mixtral-8x7B | 71.2 | 76.1 | 84.2 |
| Llama-3.1-70B | 71.0 | 68.9 | 85.2 |
| Mistral-7B | 68.8 | 65.9 | 73.9 |
| DeepSeek-V3 | 68.1 | 72.4 | 76.9 |
| Mistral-Small-24B | 63.2 | 71.8 | 79.3 |
| GPT-4o-mini | 62.7 | 73.7 | 81.9 |
| Qwen-2.5-72B | 57.1 | 60.7 | 69.7 |
| Llama-3.1-8B | 50.2 | 57.1 | 72.1 |
| **Cond. Avg** | **65.1** | **68.5** | **78.8** |

*Table 26.* Rescue rate by model in the few-shot condition on the two non-safety datasets (irony and subjectivity). Rescue rate is $P(\text{correct} \mid \text{few-shot, zero-shot wrong})$, computed pooled across the two datasets.

| Model | Rescue Rate | 95% CI | N (ZS err.) |
|---|---|---|---|
| Mixtral-8x7B | 59.9% | [53.9, 65.7] | 277 |
| Llama-3.1-8B | 52.8% | [48.3, 57.3] | 498 |
| Llama-3.1-70B | 52.8% | [46.8, 58.6] | 290 |
| GPT-4o-mini | 52.3% | [47.1, 57.4] | 373 |
| Llama-3.3-70B | 50.6% | [44.4, 56.7] | 267 |
| Mistral-Small-24B | 47.1% | [41.9, 52.4] | 367 |
| DeepSeek-V3 | 37.6% | [32.3, 43.2] | 319 |
| Qwen-2.5-72B | 27.9% | [23.6, 32.6] | 401 |
| Mistral-7B | 21.5% | [16.7, 27.0] | 260 |
| **Overall** | **45.0%** | **[43.2, 46.8]** | **3,052** |

**Stance detection: a structurally different task.**   As a further check, we ran a preliminary experiment on stance detection (SemEval-2016 Task 6; Mohammad et al. 2016) across 5 targets and 8 models. The DSF direction remains positive (partial $r = +0.295$, $N = 40$ target–model pairs). Stance detection differs structurally from our other tasks, however: it requires choosing between two explicit alternatives ("in favor" / "against") rather than classifying content against a single concept definition, which may introduce additional sources of variation beyond conceptual alignment around the definition itself. We therefore treat stance as directionally consistent but not as clean a test of the DSF–accuracy relationship as irony and subjectivity, and we report it here only as supporting evidence.

**Takeaway.**   Both core findings replicate outside the toxicity domain: DSF remains positively associated with zero-shot accuracy, and most zero-shot errors continue to resist correction even when models are given aligned definitions or few-shot examples. The effects we identify in the main paper appear to be properties of definition-driven annotation under internalized priors rather than artifacts of the safety setting.

## G. Model and Dataset Use Across Experiments

Our experiments use a two-tier structure for both models and datasets, summarized in Table 27.

The main-paper descriptive RQ2/RQ3 result tables and figures (Tables 6, 7, 8 and Figs. 3, 4) report values aggregated across all nine evaluated models (Llama-3.1-8B, Llama-3.1-70B, Llama-3.3-70B, Mistral-7B, Mistral-Small-24B, Mixtral-8x7B, DeepSeek-V3, GPT-4o-mini, Qwen-2.5-72B). All mixed-effects regression analyses (Table 5 in the main paper and Tables 11–14 in the appendix) and the multi-turn rescue analysis (Table 20) are scoped to the original six open-weights models (Llama-3.1-8B, Llama-3.1-70B, Mistral-7B, Mistral-Small-24B, Mixtral-8x7B, DeepSeek-V3) to keep statistical inference tied to the original six-model design; GPT-4o-mini, Llama-3.3-70B, and Qwen-2.5-72B contribute to the descriptive and performance tables but are excluded from the mixed-effects regressions.

The dataset side follows the same pattern. The five primary toxicity datasets (Twitter Hate, OLID, GameTox, Fox News,

Jigsaw Toxic Comments) are the analytic core and appear in every result table. The three supplementary datasets — Jigsaw Unintended Bias, SemEval Irony, Subjectivity — are used only in the specific analyses indicated below: Jigsaw Unintended Bias as a label-bias robustness check for the rescue and familiarity results, irony and subjectivity as out-of-domain replications of RQ1 and RQ2 (Appendix F).

*Table 27.* Coverage patterns used across the paper. "9 models" = all nine evaluated models (Llama-3.1-8B, Llama-3.1-70B, Llama-3.3-70B, Mistral-7B, Mistral-Small-24B, Mixtral-8x7B, DeepSeek-V3, GPT-4o-mini, Qwen-2.5-72B). "6 core" = the original six open-weights subset (Llama-3.1-8B, Llama-3.1-70B, Mistral-7B, Mistral-Small-24B, Mixtral-8x7B, DeepSeek-V3). "5 main" = the five primary toxicity datasets; Jigsaw UB = Jigsaw Unintended Bias.

| Coverage | Where it applies |
|---|---|
| 9 models × 5 main datasets | Tables 6, 7, 8; Figs. 3, 4; Table 3; Tables 15, 17, 18 (extended models shown below midrule) |
| 6 core models × 5 main datasets | Tables 5, 11–14, 16, 20, 24 |
| 9 models × 5 main + Jigsaw UB ($N{=}54$) | Tables 4, 21, 22 (familiarity / DSF) |
| 9 models × Jigsaw UB only | Table 19 (rescue-rate robustness check) |
| 9 models × Irony + Subjectivity | Tables 25, 26 (non-safety generalization) |
| Llama-3.1-8B + Mistral-7B (base) × 5 main | Table 23 (Min-K%, requires logprob access) |

