# OpenReview forum: "On the Limits of LLM Adaptability: Impact of Model-Internalized Priors on Annotation Task Performance"
_ICML.cc/2026/Conference — ICML 2026 spotlight_

### Official Review · Reviewer_KEhF · 2026-02-22

**Soundness:** 4
**Presentation:** 4
**Significance:** 3
**Originality:** 3
**Overall Recommendation:** 5
**Confidence:** 4

**Summary:**

This paper explores the extent to which a pre-trained LLM used for annotation tasks, specifically in this study’s case, toxicity, can be influenced by factors such as data familiarity, task definition (including misaligned definitions), and additional prompting information. The work introduces a new metric, Definition-Specific Familiarity (DSF) to measure the alignment between a model’s internal understanding of a concept and the specified task definition. The authors show that DSF is positively correlated with model performance on toxicity annotation tasks, whilst text memorisation does not show positive association. The findings are used to emphasise the limitations of prompt-based corrections for annotations tasks and to encourage a focus on ensuring there is task definition alignment during annotation.

**Compliance With Llm Reviewing Policy:**

Affirmed.

**Final Justification:**

The authors addressed my concerns and I have raised my confidence score accordingly.

**Key Questions For Authors:**

1. You mention that an "Enhanced DSF" variant incorporating domain-specific prompts and negative class definitions showed a weaker correlation with model performance than the primary metric used. Could you provide more intuition on why adding more granular definition alignment data reduced the metric's predictive power? A compelling explanation would strengthen confidence in the primary DSF metric’s validity.
2. Similar to Q1, have you explored any different models for the DSF metric? Is there any evidence to suggest that if a larger or different embedding space is used, would the same correlation be seen?
3. Figure 3 shows (small N) low zero-shot confidence has a similarly low rescue-rate to very high zero-shot confidence. Is the low rescue rate at low confidence a result of the model being confused by the input text itself, regardless of the definition provided, or do you have any other intuitions on what this suggests?

**Limitations:**

Yes

**Strengths And Weaknesses:**

**Strengths**
1. Metrics. The finding that text memorisation does not positively correlate to model performance and the finding that conceptual alignment matters more than memorization means that the introduction of DSF could be highly impactful for annotation tasks in future.
2. Model size & definition findings. The finding that model size non-monotonically impacts steerability for the models under analysis is a fascinating and potentially impactful finding of the study (with rescue rates for Mistral-7B around the same as Llama-3.1-70B, but lower than Llama-3.1-8B and Mistral-Small-24B). Similarly, the finding that the definition has a greater impact on accuracy variation than model selection is similarly insightful for researchers and practitioners engaging with annotation tasks.
3. Calibration findings. The finding that LLMs remain highly confident even when they are faithfully following a misaligned definition is a significant flag for practitioners.
4. Scale. The range of open weights models tested helps bolster the impacts of the findings, including a combination of MoE and dense models.

**Weaknesses**
1. Task limitation. Whilst the work presents a compelling finding in the domain of toxicity, for this work to maximise long term significance it could demonstrate the impact of DSF and the findings related to prompt steering on other tasks. This could include exploring whether these findings are safety-specific or generalise into non-safety domains.
2. DSF model selection. The DSF relies on a specific embedding model (all-MiniLM-L6-v2). It would have been beneficial to assess whether the results of the paper are sensitive to a choice of embedding models.
3. Focus on instruction-tuned models. It would be valuable to have shown whether the “decision stickiness” findings from this work results from interventions made during instruction-tuning / fine-tuning versus pre-training. By including a small number of pre-trained models for this RQ this could add to the strength of the paper’s findings.

---

> ### Author Rebuttal · Authors · 2026-03-31
>
> We sincerely thank Reviewer KEhF for the detailed and positive review. We are encouraged by the recognition of DSF as "highly impactful," the non-monotonic model size finding as "fascinating and potentially impactful," and calibration findings as "a significant flag for practitioners." We address each question below.
>
>
> **Q1. DSF sensitivity to embedding choice and scale.**
>
> We added additional analysis. DSF holds across all six embedding models tested (Table B: pairwise r > 0.90; Table A: partial r range +0.29 to +0.49). Notably, larger models tend to yield stronger partial correlations, bge-large (+0.49), e5-large (+0.43), and OpenAI (+0.44) outperform the compact MiniLM (+0.36), suggesting robustness may improve with better embeddings. Consensus DSF yields partial r = +0.41 (p = 0.003, N=54). We will adopt consensus DSF as the primary metric and add Tables A and B to Appendix E.
>
> Table A - Partial r across embedding models for DSF calculation
> |Embedding model|Partial r (Original, N=30)|Partial r (Extended, N=54)|
> |-|-|-|
> |all-MiniLM-L6-v2|+0.342|+0.363|
> |all-mpnet-base-v2|+0.358|+0.333|
> |BAAI/bge-large-en-v1.5|+0.457|+0.490|
> |intfloat/e5-large-v2|+0.430 | +0.434 |
> |hkunlp/instructor-large|+0.295|+0.301|
> |text-embedding-3-small (OpenAI)|+0.429 |+0.435|
> |**Consensus**|**+0.391**|**+0.410**|
>
> Table B - Pairwise DSF correlations (MiniLM vs. others):
> ||mpnet|bge|e5|instructor|OpenAI|
> |-|-|-|-|-|-|
> |all-MiniLM-L6-v2|0.952|0.975|0.911|0.958|0.948|
>
> **Q2. Focus on instruction-tuned models: unclear whether decision stickiness stems from instruction tuning vs. pre-training.**
>
> All evaluated models are instruction-tuned; the versions most commonly deployed for annotation in practice. As stated in our limitations (Section 5): "we cannot distinguish whether low rescue rates reflect capability limitations or intentional design choices- alignment training may deliberately limit steerability for safety reasons." The reviewer's suggestion to include base model variants is directly actionable. Given the limited rebuttal window, we were unable to complete this in time, but we plan to address it during the reviewer discussion round and camera-ready. The operationally relevant finding stands regardless: stickiness exists in the models practitioners actually use.
>
> **Q3. What explains the low rescue rate at the low-confidence end of Figure 3?**
>
> Figure 3 shows an inverted-U: rescue probability peaks at moderate confidence and declines sharply at high confidence (n>20,000), which is the most statistically reliable region, directly supporting our claim. The low rescue rate at the very low-confidence end (n<1000) is a separate phenomenon driven by context-deficient texts: these have significantly lower ROUGE-L (0.025 vs 0.061, more out-of-distribution), are dramatically shorter (median 6 vs 22 words), and contain single-word gaming fragments ("noob", "wtf", "bot") where toxicity depends on conversational context unavailable to the model. We revise the discussion to explicitly distinguish these two failure modes and annotate both in the figure.
>
>
> **Q4. Do findings generalize beyond toxicity to non-safety domains?**
>
> We chose toxicity because it is one of the most common and high-stakes LLM annotation use cases in practice [1], features competing definitions ideal for concept substitution, and our five datasets already span substantially different contexts: social media, gaming, news, forums, and civil discourse. To address generalization beyond safety, we are running a stance detection experiment (SemEval-2016 Task 6 [2]), a non-safety domain with competing definitions across five targets, directly analogous to our design. Unfortunately, running all experiments on new datasets takes substantial time via OpenRouter, results will be included as soon as available.
>
> **Q5. Why does Enhanced DSF show weaker correlation than primary DSF?**
>
> We performed additional analysis. Primary DSF outperforms the enhanced variant because it mirrors the zero-shot annotation setting: models rely on their pre-trained concept without domain scaffolding. Adding domain context causes models to produce more homogeneous descriptions, compressing between-model variance (3-20x). After controlling for dataset-level confounds, enhanced variants no longer discriminate between models (partial r drops from +0.34 to near zero, vs. raw r +0.78/+0.75). DSF is designed to answer not "is this dataset hard?" but "will this specific model align with this specific definition?", requiring between-model variation after controlling for the dataset. Primary DSF answers this; Enhanced DSF does not, becoming a dataset difficulty measure instead. This clarifies why simple elicitation works best: it captures the model's stable internal concept boundary. We will include this analysis in the revision.
>
>
>
> **References**
>
> [1] Large language model hacking: Quantifying the hidden risks of using LLMs for text annotation
>
> [2] SemEval-2016 Task 6: Detecting Stance in Tweets

---

> > ### Author Rebuttal · Reviewer_KEhF · 2026-04-01
> >
> > I appreciate the authors response, in particular the additional analysis and commitment to running experiments on non-safety domains. As a result, I have increased the confidence of my score (3 -> 4).

---

> > > ### Author Response · Authors · 2026-04-08
> > >
> > > We sincerely thank **Reviewer KEhF** for the thorough engagement with our rebuttal and for fully resolving their concerns. We are glad the additional analyses and the planned experiments on non-safety domains were well-received, and we appreciate the increased confidence score. We will ensure the revised version of the paper reflects all the improvements discussed. Thank you again for the constructive feedback that helped us strengthen the paper.
> > >
> > > **Q4. Do findings generalize beyond toxicity to non-safety domains?**
> > > - We now deliver the additional results around generalizability beyond toxicty to other domains. We ran experiments on two non-safety domains - irony detection *(SemEval-2018 Task 3; Van Hee et al., 2018)* and subjectivity detection *(Pang & Lee, 2004)* - across 9 models (original 6 plus GPT-4o-mini, Llama-3.3-70B, Qwen-2.5-72B). Both tasks use explicit human-authored definitions and binary classification, directly analogous to our toxicity setup.
> > >
> > > - DSF shows a **consistent positive association** with zero-shot accuracy across both datasets *(partial r = +0.343*, N = 18 model-dataset pairs), directionally replicating our toxicity finding *(partial r = +0.34)* and confirming the effect is not safety-specific. Decision stickiness also generalizes well, given the shift in domain: the overall few-shot rescue rate is *45.0% [43.2%, 46.8%]* across 3,052 zero-shot errors, quite similar to the 36.4% reported for toxicity (Table 6 in the paper). **Table A** summarizes accuracy by condition; **Table B** reports rescue rates.
> > >
> > > **Table A.** Accuracy (%) across conditions, averaged across irony and subjectivity datasets.
> > >
> > > | Model | Zero-Shot | Aligned Def | Few-Shot |
> > > |---|---|---|---|
> > > | Llama-3.3-70B       | 73.3 | 70.4 | 86.2 |
> > > | Mixtral-8x7B           | 71.2 | 76.1 | 84.2 |
> > > | Llama-3.1-70B       | 71.0 | 68.9 | 85.2 |
> > > | Mistral-7B              | 68.8 | 65.9 | 73.9 |
> > > | DeepSeek-V3        | 68.1 | 72.4 | 76.9 |
> > > | Mistral-Small-24B  | 63.2 | 71.8 | 79.3 |
> > > | GPT-4o-mini          | 62.7 | 73.7 | 81.9 |
> > > | Qwen-2.5-72B       | 57.1 | 60.7 | 69.7 |
> > > | Llama-3.1-8B         | 50.2 | 57.1 | 72.1 |
> > > | **Cond. Avg**        | **65.1** | **68.5** | **78.8** |
> > >
> > > **Table B.** Rescue rate by model (few-shot condition), irony and subjectivity datasets.
> > >
> > > | Model | Rescue Rate | 95% CI | N (ZS errors) |
> > > |---|---|---|---|
> > > | Mixtral-8x7B                | 59.9% | [53.9%, 65.7%] | 277 |
> > > | Llama-3.1-8B               | 52.8% | [48.3%, 57.3%] | 498 |
> > > | Llama-3.1-70B             | 52.8% | [46.8%, 58.6%] | 290 |
> > > | GPT-4o-mini               | 52.3% | [47.1%, 57.4%] | 373 |
> > > | Llama-3.3-70B            | 50.6% | [44.4%, 56.7%] | 267 |
> > > | Mistral-Small-24B        | 47.1% | [41.9%, 52.4%] | 367 |
> > > | DeepSeek-V3              | 37.6% | [32.3%, 43.2%] | 319 |
> > > | Qwen-2.5-72B             | 27.9% | [23.6%, 32.6%] | 401 |
> > > | Mistral-7B                     | 21.5% | [16.7%, 27.0%] | 260 |
> > > | **Overall**                  | **45.0%** | **[43.2%, 46.8%]** | **3,052** |
> > >
> > > We additionally ran a preliminary experiment on the originally suggested stance detection task *(SemEval-2016 Task 6; Mohammad et al., 2016)* across 5 targets and 8 models. Here as well, the DSF direction remains positive (partial r = +0.295, N = 40). We note, however, that stance detection differs structurally from our other tasks: it requires choosing between two explicit alternatives ("in favor" / "against") rather than classifying content against a single concept definition, which may introduce additional sources of variation beyond conceptual alignment around the definition itself. We therefore treat stance as directionally consistent but not as clean of a test of decision stickiness as the results we highlighted earlier. We are still pleased, that despite these differences, our approach holds. We will add all these results to the paper revision and discuss this task distinctions as well.
> > >
> > > **References:**
> > >
> > > * Van Hee et al. (2018), SemEval-2018 Task 3, https://aclanthology.org/S18-1005/
> > > * Pang & Lee (2004), ACL, https://aclanthology.org/P04-1035/
> > > * Mohammad et al. (2016), SemEval-2016 Task 6, https://aclanthology.org/S16-1003/

---

### Official Review · Reviewer_pQyU · 2026-03-12

**Soundness:** 3
**Presentation:** 3
**Significance:** 4
**Originality:** 3
**Overall Recommendation:** 5
**Confidence:** 3

**Summary:**

In this paper, the authors investigate how pre-trained priors of LLMs and user instructions affect the reliability of LLM-as-a-judge tasks. Specifically, the authors focus on the correlation between task/text familiarity and LLM annotation performance, how external knowledge or specialized prompting affects model behavior, and the role of human-written task definitions. Results indicate that Definition-Specific Familiarity (DSF), the alignment between a model's internal concept and the task definition, correlates most strongly with annotation performance. Furthermore, LLMs are sensitive to definitions in terms of under-prediction and over-prediction. Based on these results, the authors provide general advice for model and definition selection.

**Compliance With Llm Reviewing Policy:**

Affirmed.

**Key Questions For Authors:**

As noted in the limitations, the positive association between DSF and annotation performance is correlational rather than causal. It would be worth exploring whether manually crafting a definition that aligns closely with a model’s internal understanding leads to improved performance, or even directly allowing the model to generate its own task definition. Such an experiment could provide critical insights into the causal relationship between conceptual alignment and model accuracy.

**Limitations:**

yes

**Strengths And Weaknesses:**

Strengths:
1. Valuable research direction. As LLM-as-a-judge is widely adopted for annotation and evaluation, analyzing its reliability is crucial.
2. Well-structured experiments. Through three well-defined research questions, the authors conduct insightful experiments that clarify which factors most significantly impact the performance of LLM judges.
3. Practical advice for model and definition selection. The novel DSF metric provides a practical way to evaluate whether a specific model is suitable for a given task.

Weakness:
1. Although discussed in the limitations, DSF relies on similarity computation, whose robustness should be further justified. A naive experiment, such as altering the embedding model or prompts, would be super helpful.

---

> ### Author Rebuttal · Authors · 2026-03-31
>
> We sincerely thank Reviewer pQyU for the thorough and positive evaluation. We are encouraged by the recognition of our work as a "valuable research direction," the characterization of our experiments as "well-structured," and the assessment of DSF as providing "practical advice for model and definition selection." We are pleased the reviewer found the paper technically solid with excellent significance, and we address the one weakness and key question below.
>
>
> **Q1. DSF robustness: results based on a single embedding model and elicitation prompt.**
>
> We recomputed DSF across six embedding models spanning architectures and scales. DSF's positive association holds across all six (partial r range +0.29 to +0.49), in both original and extended scope (N=54, including Qwen2.5, GPT-4o-mini, Llama3.3, and Jigsaw Unintended Bias). Notably, larger models tend to yield stronger partial correlations: bge-large (+0.49), e5-large (+0.43), and OpenAI (+0.44) outperform compact MiniLM (+0.36). Consensus DSF yields partial r = +0.41 (p = 0.003, N=54), confirming robustness to embedding choice.
>
> Regarding elicitation prompt robustness for DSF, Appendix E already reports ablations across three prompt styles: Direct (neutral concept description), Expert (annotator persona), and Examples (example-free description), all variants produce positive raw correlations (DSF Simple +0.61, DSF Positive +0.78, DSF Enhanced +0.75), with Direct showing the strongest partial correlation after controlling for dataset. We will include Table A in Appendix E alongside the existing DSF variant analysis in the revision.
>
> Table A. DSF partial correlations across six embedding models.
>
> | Embedding model | Partial r (Original, N=30) | Partial r (Extended, N=54) |
> |-|-|-|
> | all-MiniLM-L6-v2 | +0.342 | +0.363 |
> | all-mpnet-base-v2| +0.358 | +0.333 |
> | BAAI/bge-large-en-v1.5 | +0.457 | +0.490 |
> | intfloat/e5-large-v2 | +0.430 | +0.434 |
> | hkunlp/instructor-large | +0.295 | +0.301 |
> | text-embedding-3-small (OpenAI) 	| +0.429 | +0.435 |
> | **Consensus (mean across 6 embeds)** 	| **+0.391** | **+0.410** |
>
>
> **Q2. Causality: does crafting definitions aligned with a model's internal concept improve performance?**
>
> We agree this is an important open question. We note, however, that RQ3 (Misalignment Susceptibility) provides quasi-causal evidence: by actively manipulating the definition provided to models - swapping aligned definitions for purposefully misaligned ones across datasets and models, varying definition scope from narrow to broad (Appendix A), and comparing zero-shot to definition-prompted conditions - we demonstrate that definition wording directly and systematically shifts model behavior (Table 7).
>
> Unfortunately, we cannot control the ground truth labels under which datasets were originally annotated, but we exploit natural variation across existing datasets as a quasi-experimental manipulation, going beyond a purely correlational design. The specific causal direction the reviewer identifies, crafting definitions to deliberately maximize DSF for a specific model, is a compelling next step, but would require new human annotations collected under the crafted definition to avoid circularity (a model will naturally score high DSF on its own generated definition). We plan to pursue this as a primary direction in future work.

---

> > ### Author Rebuttal · Reviewer_pQyU · 2026-04-04
> >
> > Thank you for the detailed response! I recommend including these details in future versions. I will keep my initial positive assessment unchanged.

---

> > > ### Author Response · Authors · 2026-04-08
> > >
> > > We sincerely thank **Reviewer pQyU** for the thorough engagement with our rebuttal and for fully resolving their concerns. We are glad the additional analyses on DSF robustness across embedding models were helpful, and we will make sure to include these details - specifically Table A with the extended DSF analysis - in the final revision. Thank you again for the constructive feedback and for maintaining the positive assessment.

---

### Official Review · Reviewer_fmDi · 2026-03-13

**Soundness:** 2
**Presentation:** 3
**Significance:** 2
**Originality:** 3
**Overall Recommendation:** 5
**Confidence:** 5

**Summary:**

# RQ:
- RQ1 (Familiarity and Performance): How does an LLM’s familiarity with the data and task definition affect annotation performance?
- RQ2 (Decision Stickiness): To what extent can external knowledge or specialized prompting correct initial zero-shot errors, and how does
model confidence relate to correctability?
- RQ3 (Misalignment Susceptibility): How do LLMs behave when given misaligned or incorrect task definitions, and can confidence scores detect such misalignment?

# Contribution:
- Found that nearly two-thirds of zero-shot errors are resistant to correction via prompting
- Demonstrated that conceptual alignment (DSF) is a much stronger predictor of performance (partial $r=+0.34$) than text-level memorization/contamination (partial $r=-0.19$).
- Revealed that LLMs remain highly confident even when following incorrect or misaligned definitions,
- specific wording of a definition often causes larger performance swings than the choice of the model itself. (prompt sensitivity)

**Compliance With Llm Reviewing Policy:**

Affirmed.

**Final Justification:**

Q2-4 are resolved.
Q1, due to time restraints is a promise.
Will move to accept -- , please include answers to Q1.

**Key Questions For Authors:**

# Questions
1. Toxicity is fundamentally subjective, and recent work (e.g., from the Workshop on Online Abuse and Harms) suggests that "errors" in LLM classification often represent valid alternative interpretations rather than objective failures.
  - If a significant portion of your "unrecoverable errors" are actually cases of human-labeler disagreement or ambiguity, does this change your interpretation of "decision stickiness" as a model limitation versus a reflection of task inherent noise?
  - Try more datasets from toxicity (more recent) or other domains where the classification isn't as subjective

2. Jigsaw is known to have significant issues with unintended bias (e.g., flagging African American Vernacular English as toxic) which led to the creation of more robust versions like the Jigsaw Unintended Bias dataset. How do your results on "Rescue Rates" change when using datasets explicitly designed to minimize identity bias?

3. Did you experiment with iterative / multi-turn prompting to "break" the stickiness of high-confidence errors, and if so, did the 36.4% rescue rate improve?

4. To what extent do the correlations between DSF and performance hold when using state-of-the-art, larger embedding models or different similarity measures (e.g., Mahalanobis distance)?

## Note
- Addressing  Q2, Q3, Q4 ==> weak accept
-  additional addressing Q1 adequately (esp with other domain)  ==> accept

**Limitations:**

Toxicity & bias limitations ... lots of literature not discussed

**Strengths And Weaknesses:**

# Soundness
## Strength
- Definition-Specific Familiarity (DSF) to isolate "conceptual alignment" from "textual memorization,"
- 6 diverse LLMs (dense and MoE) across 5 distinct datasets.
- mixed-effects logistic regression, under 10 different prompting conditions.
- Swapping definitions across domains provides empirical evidence for how LLMs shift decision thresholds while revealing a significant calibration failure
## Weakness
- only one embedding model `all-MiniLM-L6-v2`
- toxicity is subjective --> dataset ground truth is debatable
  - (look into Workshop on Online Abuse & Harms, many many papers on toxicity & subjectivity)
  - jigsaw is out of date, there's an unbiased version published later)
  - https://hatespeechdata.com/
  - https://github.com/aymeam/Datasets-for-Hate-Speech-Detection (simple google search)
- different models (e.g., Llama vs. DeepSeek) often require different prompt structures to perform optimally.
- w/o fine-tuning a model to intentionally shift its internal concept (DSF) while holding other variables constant, the paper cannot definitively prove that pre-trained priors cause the lack of steerability.
- the findings are limited to binary classification in a single domain (toxicity).

# Presentation
## Strength
- logical, clear visual (Fig 1), odds ratio & partial correlations
## Weakness
- a table to summarize the metrics
- missing prompt in the appendix
- appendix shows CONDA definitions for the game toxicity (dota 2 dataset) but tests on the GameTox which is on World of Tanks. CONDA dataset is also open source, so why not use that?

# Significance:
- impactful, as people do zero-shot w/ LLMs
- can show limit of prompt engineering
- methodological blueprint for "stress-testing" LLMs by using concept substitution

# originality
## Strength
-reframes LLM performance as a struggle between pre-trained "anchors" and user instructions.
- DSF metric
- RQ3
- 63.6% of zero-shot errors are unrecoverable via prompting
## weakness
- overlap with steerability literature
- lots of dataset is on toxicity, but are using outdated datasets (e.g., jigsaw unbiased vs jigsaw). Toxicity is also very subjective
- no proposed solution

---

> ### Author Rebuttal · Authors · 2026-03-31
>
> We sincerely thank Reviewer fmDi for the thorough review. We are encouraged by the recognition of DSF's ability to "isolate conceptual alignment from textual memorization," the characterization of our approach as a "methodological blueprint for stress-testing LLMs," and the clear path to acceptance outlined. We address each of Q1, Q2, Q3, Q4 directly below.
>
> **Q2. Toxicity ground truth is subjective; Jigsaw has known unintended bias issues.**
>
> We acknowledge that toxicity is inherently subjective. Our study does not treat any annotation scheme as ground truth. Instead, each dataset provides an explicit human-authored definition, and our analysis measures LLM alignment with that policy. Our five datasets span multiple domains, temporal periods, and annotator teams, making systematic bias in a single direction unlikely. Regarding Jigsaw, we ran our full rescue rate analysis on the Jigsaw Unintended Bias [3] and find results remain consistent (table below), confirming decision stickiness is not an artifact of Jigsaw's known limitations.
>
> |Model|Rescue Rate|95% CI|N (ZS errors)|
> |-|-|-|-|
> |Mistral-Small-24B|41.9%|[39.9, 44.0]|2,248|
> |DeepSeek-V3|40.8%|[38.7, 42.9]|2,112|
> |Llama-3.1-8B|37.9%|[35.9, 39.9]|2,232|
> |Llama-3.1-70B|34.1%|[32.2, 36.1]|2,296|
> |Mixtral-8x7B|31.8%|[29.9, 33.8]|2,232|
> |Mistral-7B|28.9%|[27.0, 30.8]|2,144|
>
> **Q3. Does iterative / multi-turn prompting improve the 36.4% rescue rate?**
>
> Following the Self-Refine framework [1], we ran an exploratory multi-turn experiment sampling 100 zero-shot errors per model x dataset pair. Each error underwent a 3-turn sequence: Turn 1 added few-shot examples, Turn 2 added the aligned definition, Turn 3 asked the model to reconsider with full response history. Overall rescue reached 7.5%, 16.3%, and 18.7% across turns; high-confidence errors reached only 2.0%, 7.1%, and 8.5%. The largest gain came at Turn 2, suggesting that even in an iterative setting, conceptual alignment matters more than repeated correction attempts. Notably, even when shown their own prior responses and confidence scores, models tended to stay anchored to earlier judgments rather than revise them, consistent with [2]. Directionally, results remained well below the paper's single-turn rate (36.4%; note: different sample sizes), confirming decision stickiness is robust to iterative correction.
>
> |Model|R@1|R@2|R@3|
> |-|-|-|-|
> |DeepSeek-V3|8.2%|20.8%|27.4%|
> |Llama-3.1-70B|10.6%|14.6%|19.0%|
> |Mistral-7B|3.0%|12.3%|12.7%|
> |Mistral-Small-24B|7.0%|12.4%|12.9%|
> |Mixtral-8x7B|5.8%|17.1%|18.5%|
> |**Overall**|**7.5%**|**16.3%**|**18.7%**|
> |**Overall (high-conf.)**|**2.0%**|**7.1%**|**8.5%**|
>
> **Q4. DSF relies on a single embedding model, does it hold across larger or different models?**
>
> We recomputed DSF across six embedding models spanning architectures and scales. DSF's positive association with accuracy holds across all six (partial r range +0.29 to +0.49), in both original and extended scope (N=54, including Qwen2.5, GPT-4o-mini, Llama3.3, and Jigsaw Unintended Bias). Consensus DSF yields partial r = +0.41 (p = 0.003, N=54). We will adopt consensus DSF as the primary metric in the revision.
>
> Table A. DSF partial correlations across six embedding models.
> |Embedding model|Partial r (Original, N=30)|Partial r (Extended, N=54)|
> |-|-|-|
> |all-MiniLM-L6-v2|+0.342|+0.363|
> |all-mpnet-base-v2|+0.358|+0.333 |
> |BAAI/bge-large-en-v1.5|+0.457|+0.490|
> |intfloat/e5-large-v2|+0.430|+0.434|
> |hkunlp/instructor-large| +0.295 | +0.301 |
> |text-embedding-3-small (OpenAI)|+0.429|+0.435|
> |**Consensus (mean across 6 embeds)**|**+0.391**|**+0.410**|
>
> Table B. Pairwise DSF correlations (MiniLM vs. other models):
> ||mpnet|bge|e5|instructor|OpenAI|
> |-|-|-|-|-|-|
> |MiniLM|0.952|0.975|0.911|0.958|0.948|
>
> **Q1. Findings limited to toxicity and binary classification, do they generalize?**
>
> We chose toxicity because it is one of the most common and high-stakes LLM annotation use cases in practice [4], features competing definitions across datasets ideal for concept substitution, and binary classification is the dominant annotation format. To address generalizability, we are running a preliminary experiment on stance detection (SemEval-2016 [5]), a non-safety domain with competing definitions across five targets, directly analogous to our concept substitution design. Results will be included in the revision.
>
>
>
> Unfortunately, due to length limits, please kindly refer to **Q4 & 5** under reviewer sMTu for our replies to prompt optimization per model and causal attribution of stickiness.
>
>
>
> **References:**
>
> [1] Self-Refine: Iterative Refinement with Self-Feedback
>
> [2] When Can LLMs Actually Correct Their Own Mistakes?
>
> [3] https://huggingface.co/datasets/google/civil_comments
>
> [4] Large language model hacking: Quantifying the hidden risks of using LLMs for text annotation
>
> [5] SemEval-2016 Task 6: Detecting Stance in Tweets

---

> > ### Author Rebuttal · Reviewer_fmDi · 2026-04-03
> >
> > Q2-4 are resolved.
> > Q1, due to time restraints is a promise.
> > Will move to accept -- , please include answers to Q1.

---

> > > ### Author Response · Authors · 2026-04-08
> > >
> > > We sincerely thank **Reviewer fmDi** for moving to *Accept* and for the clear path outlined. We are pleased to now deliver additional results on Q1. Our experiments on irony and subjectivity detection confirm that both DSF's positive association with accuracy and decision stickiness replicate beyond toxicity.
> > >
> > > **Q1. Generalizability beyond toxicity.**
> > >   - We ran experiments on two non-safety domains - irony detection (SemEval-2018 Task 3; Van Hee et al., 2018) and subjectivity detection (Pang & Lee, 2004) - across 9 models (original 6 plus GPT-4o-mini, Llama-3.3-70B, Qwen-2.5-72B). Both tasks use explicit human-authored definitions and binary classification, directly analogous to our toxicity setup.
> > >
> > >   - DSF shows a **consistent positive association** with zero-shot accuracy across both datasets *(partial r = +0.343*, N = 18 model-dataset pairs), directionally very similar to our toxicity finding *(partial r = +0.34)*. Decision stickiness also generalizes: the overall few-shot rescue rate is *45.0% [43.2%, 46.8%]* across 3,052 zero-shot errors, with a majority again resisting correction (analogous to Table 6 in the paper). The replication across a substantial domain shift reinforces that limited error recovery is a general property of LLM annotation, not a toxicity-specific artifact. **Table A** summarizes accuracy by condition; **Table B** reports rescue rates.
> > >
> > > **Table A.** Accuracy (%) across conditions, averaged across irony and subjectivity datasets.
> > >
> > > | Model | Zero-Shot | Aligned Def | Few-Shot |
> > > |-|-|-|-|
> > > | Llama-3.3-70B            | 73.3 | 70.4 | 86.2 |
> > > | Mixtral-8x7B               | 71.2 | 76.1 | 84.2 |
> > > | Llama-3.1-70B            | 71.0 | 68.9 | 85.2 |
> > > | Mistral-7B                    | 68.8 | 65.9 | 73.9 |
> > > | DeepSeek-V3              | 68.1 | 72.4 | 76.9 |
> > > | Mistral-Small-24B       | 63.2 | 71.8 | 79.3 |
> > > | GPT-4o-mini                | 62.7 | 73.7 | 81.9 |
> > > | Qwen-2.5-72B            | 57.1 | 60.7 | 69.7 |
> > > | Llama-3.1-8B             | 50.2 | 57.1 | 72.1 |
> > > | **Cond. Avg**            | **65.1** | **68.5** | **78.8** |
> > >
> > > **Table B.** Rescue rate by model (few-shot condition), irony, and subjectivity datasets.
> > >
> > > | Model | Rescue Rate | 95% CI | N (ZS errors) |
> > > |-|-|-|-|
> > > | Mixtral-8x7B             | 59.9% | [53.9%, 65.7%] | 277 |
> > > | Llama-3.1-8B           | 52.8% | [48.3%, 57.3%] | 498 |
> > > | Llama-3.1-70B         | 52.8% | [46.8%, 58.6%] | 290 |
> > > | GPT-4o-mini             | 52.3% | [47.1%, 57.4%] | 373 |
> > > | Llama-3.3-70B         | 50.6% | [44.4%, 56.7%] | 267 |
> > > | Mistral-Small-24B    | 47.1% | [41.9%, 52.4%] | 367 |
> > > | DeepSeek-V3         | 37.6% | [32.3%, 43.2%] | 319 |
> > > | Qwen-2.5-72B           | 27.9% | [23.6%, 32.6%] | 401 |
> > > | Mistral-7B               | 21.5% | [16.7%, 27.0%] | 260 |
> > > | **Overall**              | **45.0%** | **[43.2%, 46.8%]** | **3,052** |
> > >
> > >   - We additionally ran a preliminary experiment on the originaly suggested stance detection (SemEval-2016 Task 6; Mohammad et al., 2016) across 5 targets and 8 models. DSF direction remains positive (partial r = +0.295, N = 40), we note, however, that stance detection differs structurally from our other tasks: it requires choosing between two explicit alternatives ("in favor" / "against") rather than classifying content against a single concept definition, which may introduce additional sources of variation beyond conceptual alignment around the definition itself. We therefore treat stance as directionally consistent but not as clean a test of decision stickiness as results from irony and subjectivity detection we highlighted earlier. Yet, even in this setting, the value of DSF still holds. We will incorporate these results and add a distinction of this in the revision.
> > >
> > > **References:**
> > >
> > > * Van Hee et al. (2018), SemEval-2018 Task 3, https://aclanthology.org/S18-1005/
> > > * Pang & Lee (2004), ACL, https://aclanthology.org/P04-1035/
> > > * Mohammad et al. (2016), SemEval-2016 Task 6, https://aclanthology.org/S16-1003/

---

### Official Review · Reviewer_sMTu · 2026-03-13

**Soundness:** 3
**Presentation:** 2
**Significance:** 2
**Originality:** 3
**Overall Recommendation:** 5
**Confidence:** 3

**Summary:**

This paper considers how LLM's pre-trained "concepts" collide with prompts attempting to steer them. The paper introduces DSF, a metric to measure alignment between a model's internal concept and a task definition. This metric predicts performance better than text memorization. Most zero-shot errors resist correction via prompt, especially when the model self reports high confidence--- confidence is not a good way to detect misalignment. The authors fine definition design matters more than model choice or prompt engineering for these annotation tasks.

**Compliance With Llm Reviewing Policy:**

Affirmed.

**Final Justification:**

My main concerns have been address--- the authors have provided strong evidence for their method during the rebuttal period. The only remaining suggestions are around framing.

I believe this paper meets the bar for acceptance.

**Key Questions For Authors:**

See weaknesses.

**Limitations:**

Yes.

**Strengths And Weaknesses:**

Strengths:

The core question is decently motivated. DSF is simple, intuitive, and easy to compute.

Weaknesses:

1. The usage of ROUGE-L is a bit odd. It feels like a weak baseline. In this age, ROUGE-L is a weak proxy for memorization. The authors even cite Min-K% Prob from Shi et.al 2024, but don't use it. Some perplexity based or likelihood based metric would be more convincing.

2. The model selection (table 3) is quite outdated. There are newer base models of similar size which have been available for a while now (e.g. qwen models).

3. Figure 3 doesn't really support the narrative. It is written that "high confidence errors are the hardest to correct" but the graph literally shows the hardest errors to correct and the lowest confidence errors.

4. No comparison against automated prompt optimization (e.g. DSPy). This seems relevant, though I'm not sure on the usage of these auto optimizers on base models.

5. I'm confused on the "pre-training" narrative. Are the models testing pre-trained bases or instruct models?
    a. If the former: why pre-trained LLMs are the focus here. Can't "concepts" be ingrained at any training point? Are instructed tuned LLMs what we use for these annotation tasks anyway?
    b. If the latter: how can be attribute the source of these sticky concepts to the pre-training? Couldn't it be from instruction tuning as well? The fixation on pre-training feels misdirected.

6. The DSF correlation is modest. Especially against the weak ROUGE-L baseline, I don't think it is amazingly convincing of a result.

---

> ### Author Rebuttal · Authors · 2026-03-31
>
> We sincerely thank Reviewer sMTu for the careful reading and constructive feedback. We are encouraged by the recognition that the core question is fairly well motivated and that DSF is simple, intuitive, and easy to compute. We address each question below.
>
> **Q1. ROUGE-L is a weak memorization baseline; Min-K% Prob preferred. DSF correlation is modest.**
>
> Min-K% Prob requires per-token log probabilities, which were unavailable for most frontier API models we tested via OpenRouter, Together AI, and HuggingFace inference, limiting its applicability to black-box annotation pipelines. In addition, Min-K%-style scores are model-specific and depend on model-dependent probability scales and a tuned choice of k, making them less suitable for cross-model comparisons. ROUGE-L, while simple, is computable for any model and fully model-agnostic. On the ≤ 8b LLMs where logprobs were available (N=2 models x 5 dataset pairs), Min-K% and ROUGE-L tell the same story in our data: both are negative, while DSF is positive (Table B). The DSF effect is also non-trivial: partial r=+0.34 is medium by Cohen [1] conventions, rises to r=+0.66 in few-shot settings, and holds across 6 embedding models (Table A; Reviewer fmDi response).
>
> Table B. Mixed-effects: Accuracy ~ metric + (1|dataset) + (1|model)
>
> |Metric|N|Raw Coef.|Std. Coef.|p|
> |-|-|-|-|-|
> |ROUGE-L|10|-2.479|-0.755|p=0.001|
> |Min-K%|10|-0.044|-0.596|p=0.036|
> |DSF|10|+0.425|+0.838|p<0.001|
>
> **Q2. Some Table 3 models are outdated. Include newer ones (e.g., Qwen)**
>
> We expanded evaluation to include GPT-4o-mini, Llama-3.3-70B, and Qwen-2.5-72B (directly addressing this concern), plus Jigsaw Unintended Bias as an additional dataset. Rescue rates for all new models remain well below 50%, and DSF partial correlations in the extended setting (N=54) remain significant (p < 0.001), confirming findings generalize to newer model families.
>
> Table 3 (new model rows):
> |Model|Zero-Shot|Aligned Def|Few-Shot|FS + Def|Misaligned|Overall|
> |-|-|-|-|-|-|-|
> |GPT-4o-mini|81.6/.871|83.3/.875|84.1/.880|83.3/.880|81.1/.857|82.3/.862|
> |Llama-3.3-70B|80.5/.816|82.3/.815|82.0/.821|83.0/.826|79.0/.769|80.1/.795|
> |Qwen-2.5-72B|83.3/.809|82.2/.792|83.8/.894|83.2/.892|81.2/.775|82.2/.796|
>
> Table 6 (extended):
> |Model|Rescue Rate|95% CI|N (ZS errors)|
> |-|-|-|-|
> |GPT-4o-mini|35.7%|[34.6, 36.8]|7,152|
> |Llama-3.3-70B|30.4%|[29.4, 31.4]|7,592|
> |Qwen-2.5-72B|27.8%|[26.8, 28.9]|6,504|
>
>
> **Q3. Figure 3 does not support the narrative, i.e., the hardest errors to correct seem to be the lowest confidence**
>
> Figure 3 shows an inverted-U: rescue probability peaks at moderate confidence and declines sharply at high confidence (n>20,000), which is the most statistically reliable region, directly supporting our claim. The low rescue rate at the very low-confidence end (n<1000) is a separate phenomenon driven by context-deficient texts: these have significantly lower ROUGE-L (0.025 vs 0.061, more out-of-distribution), are dramatically shorter (median 6 vs 22 words), and contain single-word gaming fragments ("noob", "wtf", "bot") where toxicity depends on conversational context unavailable to the model. We revise the discussion to explicitly distinguish these two failure modes and annotate both in the figure.
>
> **Q4. No comparison against automated optimizers (e.g., DSPY)**
>
> We ran DSPy [2], automated prompt optimization under two conditions: DSPy Optimized (optimized prompt examples) and DSPy Aligned (optimized examples with aligned definition). DSPy yields average accuracies of 79.5% and 80.8%, respectively. These are very close to the simple aligned definition baseline (81.7%), confirming that the limits we observe reflect pre-trained concept anchoring rather than suboptimal prompt design. Full results added to Table 3.
>
> ||0-Shot|Aligned Def|Few-Shot|FS + Def|Misaligned|DSPy Aligned|DSPy Optimized|Overall|
> |-|-|-|-|-|-|-|-|-|
> |**Cond. Avg**|79.5/.829|81.7/.822|80.6/.842|80.9/.832|79.9/.761|80.8/.803|79.5/.794|80.1/.770|
>
>
> **Q5. Confusion around the use of base or pretrained models, as well as the attribution of stickiness to different training steps, needs clarification.**
>
> All evaluated models are instruction-tuned, which are the versions most commonly deployed for annotation in practice. As stated in our limitations (Section 5): "we cannot distinguish whether low rescue rates reflect capability limitations or intentional design choices - alignment training may deliberately limit steerability for safety reasons." We extend this to DSF: we deliberately avoid causal claims, positioning DSF as a diagnostic metric regardless of which training stage produces concept alignment. The operationally relevant finding is that stickiness exists in the models practitioners actually use, regardless of training-stage origin. We plan to include base model variants before camera-ready.
>
> **References**
>
> [1] Statistical Power Analysis for the Behavioral Sciences
>
> [2] DSPy: Compiling Declarative Language Model Calls into Self-Improving Pipelines

---

> > ### Author Rebuttal · Reviewer_sMTu · 2026-04-04
> >
> > **Q1:** Mostly addressed. The black-box api constraint is valid. Though, I do think other methods would still work despite this constraint while being a bit more robust compared to ROUGE. e.g. embedding based similarity, BERTScore, etc.
> >
> > **Q2:** Fully addressed. Thank you for including these new models!
> >
> > **Q3:** Reasonable clarifying answer. Revising will help clear up the confusion greatly.
> >
> > **Q4:** Addressed. Thank you for running this experiment.
> >
> > **Q5:** Not well addressed. Though, I don't see this as a core issue, it's a framing issue. I don't understand why the authors frame the narrative around "pre-trained priors"--- there's no evidence the priors come from the pre-training stage. Now, that doesn't mean they don't exist, the experiments/method clearly measure/do something, it's just not necessarily from pre-training. To be honest, many of these "pre-trained priors" and probably more likely to come from instruction/RLHF tuning, e.g. notions of what is toxic/harmful. I think a lot of my confusion with this paper was trying to understand where pre-training fit in when it really isn't a necessary part of the story.
> >
> > **Q6:** Unaddressed.
> >
> > The authors have shown solid evidence around my questions. In particular, Q2 and Q4. The explanation on Q3 clears up a relatively big confusion/concern. While I recommend the authors revisit the paper framing, that is outside the core findings and methods, in my opinion.
> >
> > As such, I will raise my score from 3 to 4.

---

> > > ### Author Response · Authors · 2026-04-06
> > >
> > > We thank the reviewer for the careful re-read and the score increase. We address the remaining points.
> > >
> > > * **Q1 (remaining). Other methods would still work despite this constraint and are more robust than ROUGE. e.g. embedding-based similarity, BERTScore, etc.**
> > >
> > > **Response:** We additionally computed BERTScore and embedding cosine similarity, the reviewer's suggested alternatives, as more semantically sensitive memorization baselines. Both use the same all-MiniLM-L6-v2 encoder as DSF, ensuring any difference reflects signal type rather than encoder choice. The extended Table 4 below shows that all three memorization metrics show negative partial correlations with accuracy after controlling for dataset, while DSF remains positive. The entire family of text memorization metrics, lexical, contextual, and semantic, points in the wrong direction, confirming that the issue is not ROUGE-L's sensitivity as a memorization proxy, but that text memorization itself does not predict annotation performance.
> > >
> > > | Metric | Raw r | Partial r |
> > > |---|---|---|
> > > | Text Memorization (ROUGE-L) | −0.80 | −0.19 |
> > > | BERTScore | −0.76 | −0.15 |
> > > | Embedding Similarity | −0.71 | −0.16 |
> > > | Definition Familiarity (DSF) | +0.61 | +0.34 |
> > >
> > > * **Q5 (remaining). "Pre-trained priors" framing implies a training-stage attribution our experiments cannot establish.**
> > >
> > > **Response:** The reviewer is correct, and we concede this point directly. Our use of "pre-trained priors" may indeed be interpreted as implying attribution to a specific training stage that our experiments cannot establish. Instruction tuning and RLHF/DPO are equally plausible, and possibly more likely, sources of internalized concepts like toxicity. As the reviewer kindly noted, this is a framing issue, not a methods issue: the core findings hold regardless of which training stage produced the observed concept anchoring.
> > > In the revision, we will replace "pre-trained priors" with "model-internalized priors" throughout, revise the narrative framing accordingly, and explicitly acknowledge in the limitations that our design cannot distinguish between training stages as sources of concept anchoring.
> > >
> > > * **Q6 (remaining). The DSF correlation is modest. Especially against the weak ROUGE-L baseline, I don't think it is amazingly convincing of a result.**
> > >
> > > **Response:** We thank the reviewer for pressing on this; it deserves a direct and complete response. In our original rebuttal, length constraints led us to address Q6 jointly with Q1. Here we respond fully, extending beyond the original emphasis on partial r=+0.34 being medium by Cohen  [1] conventions.
> > >
> > > - **Direction, not just magnitude.**  As shown in the extended Table 4 (Q1), all four memorization metrics: ROUGE-L (r=−0.19), BERTScore (r=−0.15), and embedding similarity (r=−0.16) show negative partial correlations while DSF remains positive (+0.34), stable across N=30 and N=54. The difference is one of direction, not only magnitude. Translating to accuracy points: ROUGE-L's full range corresponds to less than 1 accuracy point in the wrong direction, while DSF's full range corresponds to approximately 21 points in the right direction.
> > >
> > > - **Practical utility.** DSF requires no labeled data, only the task definition and a handful of prompts. Its association with accuracy is consistent across prompting conditions: r=+0.34 (zero-shot), r=+0.37 (aligned definition), and r=+0.66 (few-shot). The few-shot amplification likely arises because models with stronger conceptual alignment better leverage examples, not because DSF incorporates them. The partial r remains positive when any single dataset is excluded, confirming the result is not driven by one outlier. Each 0.1-unit increase in DSF corresponds to approximately 0.8 accuracy points within a dataset after controlling for dataset difficulty, with the within-dataset accuracy gap between best and worst model averaging 7.6 points (zero-shot) and 9.3 points (few-shot). Crucially, this gap is not simply explained by model size: steerability is not monotonic in model size (Table 6), meaning DSF provides a practical pre-screening signal proportional to real annotation performance differences.
> > >
> > > - **The negative memorization result is not counterintuitive.** The raw r=−0.80 is driven by a dataset confound: Jigsaw is simultaneously the highest-memorization (ROUGE-L=0.084) and hardest dataset (accuracy=0.705); GameTox has near-zero memorization (0.001) yet the highest accuracy (0.902). Controlling for dataset difficulty reduces this to a small residual (partial r=−0.19). Crucially, neither Shi et al. nor Golchin & Surdeanu evaluate memorization as a predictor of annotation performance - both measure contamination detection. Our paper is the first to ask this question directly.
> > >
> > > We will include these clarifications and new experiments in the final paper. We hope they address the remaining concerns, and we would be grateful if you would consider raising the score if these points feel better addressed.

---

### Decision · Program_Chairs · 2026-04-30

**Decision:**

Accept (spotlight)

**Comment:**

This paper studies the limits of prompt-based steerability in LLM annotation tasks and introduces Definition-Specific Familiarity (DSF) as a measure of alignment between model-internal concepts and task definitions .

Reviewers consistently find the problem important and the experimental analysis thorough, with DSF providing a simple yet effective signal for predicting annotation performance. The work offers strong empirical evidence that many zero-shot errors are resistant to correction and that definition alignment plays a more critical role than memorization. The rebuttal further strengthens the paper by adding experiments on newer models, additional embedding analyses, and validation beyond toxicity, addressing most concerns.

While some limitations remain regarding framing and domain scope, the overall contribution is clear, well-supported, and practically relevant.

I recommend acceptance.